# Morphological variability within the indigenous sheep population of Benin

Habib Rainier Vihotogbe Whannou[1], Cossi Ulriche Afatondji[1], Maurice Cossi Ahozonlin[1], Martin Spanoghe[2], Deborah Lanterbecq[2], Dominique Demblon[2], Marcel Romuald Benjamin Houinato[1], Luc Hippolyte Dossa[1]*

1 Ecole des Sciences et Techniques de Production Animale, Faculté des Sciences Agronomiques, Université d'Abomey-Calavi, Abomey-Calavi, Bénin, 2 Département Agro-biosciences et Chimie, Haute Ecole Provinciale de Hainaut (HEPH) Condorcet, Ath, Belgique

* hippolyte.dossa@fsa.uac.bj

**Data Availability Statement:** All relevant data are within the manuscript and its Supporting Information files.

## Abstract

Knowledge of both the genetic diversity and geographical distribution of animal genetic resources is a prerequisite for their sustainable utilization, improvement and conservation. The present study was undertaken to explore the current morphological variability within the sheep population in Benin as a prelude for their molecular characterization. From November 2018 to February 2020, 25 quantitative linear body measurements and 5 qualitative physical traits were recorded on 1240 adult ewes from the 10 phytogeographic zones that comprise the three vegetation zones of Benin. Fourteen morphological indices were calculated based on the linear body measurements. The collected data were first analyzed using multiple comparisons of least-square means (LSmeans), followed by generalized linear model (GLM) procedures, to explore the relationships among the measured morphometric traits and the 10 phytogeographic zones. Next, the presence of any genetic sub-populations was examined using multivariate analytical methods, including canonical discriminant analysis (CDA) and ascending hierarchical clustering (AHC). Univariate analyses indicated that all quantitative linear body measurements varied significantly (P<0.05) across the phytogeographic zones. The highest values (LSmean± standard error) of withers height (68.3±0.47 cm), sternum height (46.0±0.35 cm), and rump height (68.8±0.47 cm) were recorded in the Mekrou-Pendjari zone, the drier phytogeographic zone in the North, whereas the lowest values, 49.2±0.34, 25.9±0.26, and 52.0±0.35 cm, respectively, were recorded in the Pobe zone in the South. Multivariate analyses revealed the prevalence of four distinct sheep sub-populations in Benin. The sub-population from the South could be assimilated to the short-legged and that from the North to the West African long-legged sheep. The two other sub-populations were intermediate and closer to the crossbreeds or another short-legged sub-breed. The proportion of individuals correctly classified in their group of origin was approximately 74%. These results uncovered a spatial morphological variation in the Beninese sheep population along a South-North phytogeographic gradient.

**Funding:** This work is financially supported by the Government of Belgium through the Académie de Recherche et d'Enseignement supérieur (ARES). ARES-PRD Project entitled "Amélioration des systèmes traditionnels d'élevage de petits ruminants (ovins et caprins) dans un contexte de mutation environnementale et sociétale au Bénin". https://www.ares-ac.be/en/cooperation-au-developpement The funders had no role in study design, data collection and analysis, decision to publish, or preparation of the manuscript.

**Competing interests:** The authors have declared that no competing interests exist.

## Introduction

Small ruminants play important socio-economic and cultural roles [1, 2] and contribute to improved livelihoods in both rural and peri-urban areas [3, 4]. For millions of rural households, the holding of these animal genetic resources, notably sheep, represents a pathway to poverty reduction and an increase in financial security [2, 5, 6]. Furthermore, they make an important contribution to the protein consumption of smallholder households [7]. In West Africa, sheep populations are raised under harsh and diverse ecological conditions, which may have led to the evolution of diversified adaptive traits for their survival [2, 8]. This valuable diversity of livestock is increasingly exposed to socio-economic and ecological changes that threaten their genetic integrity. Indeed, changes in production systems, breed preferences of farmers and market demands are the main drivers of their genetic dilution through poorly planned or indiscriminate crossbreeding, while their potential for further genetic improvements remains largely unknown.

In Benin, as in most West African countries, indigenous sheep populations are not sufficiently characterized, and little reliable data are available [9]. It is commonly accepted that two main sheep breeds, the Djallonké (S1 and S2 Figs) and the Sahelian (S3 and S4 Figs), are widely distributed throughout Benin [10–12]. Djallonké, also named West African Dwarf (WAD) sheep, seems to originate from the breed of fine-tailed and hairy sheep native to Western Asia, having migrated to Africa through the Isthmus of Suez and Bab el Mandeb. Moreover, Djallonké sheep were the only sheep breed in the African continent until the third millennium BC [13]. Widely distributed in West Africa, Djallonké sheep are mainly raised for meat [10, 11, 14]. Moreover, Djallonké sheep are particularly adapted to coastal areas [15] because of their resistance to trypanosomiasis [12, 16]. However, Djallonké sheep may have undergone significant phenotypic changes over time [9, 12]. Generally, Djallonké/WAD sheep are small-sized animals with straight facial profiles, small narrow-erected ears, and a hairy short coat [14, 17]. In contrast to ewes, rams are horned and have a heavy white or pied mane black forequarters and white hindquarters. Two sub-breeds of Djallonké sheep have been identified based on size [18, 19]: the larger breed is found in the Sudanian zone, and the smaller breed in the Guinean zone further south [9, 18].

Sahelian sheep include all long-legged sheep breeds known under different ethnic and local names in the semi-arid and arid zones of the West African Sahel [9]. Similar to Djallonké sheep, Sahelian sheep are thought to have descended from the fine-tailed and hairy sheep [9]. Although Sahelian sheep are not known to survive in humid areas [20], they are increasingly encountered in different humid localities of West Africa, including Benin, over the past few years [11, 12], reflecting their progressive adaptation to less dry climates. Sahelian sheep are raised for meat and milk production in Sahelian pastoral and agro-pastoral production systems [14]. Sahelian sheep have a convex facial profile, long pendulous ears, a long thin tail, and diverse coat color [9]. A typical characteristic of the Sahelian ram is the absence of mane. As in several West African countries, many crossbreeds between Sahelian and Djallonké sheep (S5 and S6 Figs) are present in Benin with various intermediate body sizes.

The lack of knowledge on the genetic diversity of West African sheep populations and their specific traits constitutes a major constraint for implementing sound programs for their genetic improvement and sustainable use. Moreover, the presence of unknown sub-breeds within each of these two known breeds of sheep and the occurrence of crossbreeding can lead to certain ambiguities when it comes to distinguishing certain individuals according to well-defined breed/genetic type standards. According to the Food and Agriculture Organization Agency (FAO) [21], phenotypic and molecular characterizations are important tools for documenting breeds, which is the first step towards the development of strategies for their

sustainable use, management and conservation. To date, neither of these characterization tools have not been covered in depth to describe the diversity existing within the Beninese sheep population. Hence, in this study, to further document this existing diversity and to explore the actual spatial distribution within the indigenous sheep population of Benin, we primarily characterized their morphology based on a large panel of collected morphological/phenotypic traits. We hypothesized that the sheep population of Benin is highly diverse and unevenly distributed according to ecological conditions.

The current study aimed to establish the relationships among sheep morphometric traits and the 10 phytogeographic zones of Benin using univariate analyses and then explore the presence of sheep sub-populations in the Beninese indigenous sheep population using multivariate analyses. The findings of this study will provide the basis for a comprehensive molecular study on the same samples, based on both simple sequence repeat and single nucleotide polymorphism marker genotyping. Morphological data could then be compared with molecular data and association analyses (i.e., genome-wide association studies) to appropriately address possible breeding strategies for the indigenous sheep population of Benin.

## Material and methods

### Ethical statement

This study was conducted as part of the ARES-PRD Project (Amélioration des systèmes traditionnels d'élevage de petits ruminants (ovins et caprins) dans un contexte de mutation environnementale et sociétale au Bénin https://www.ares-ac.be/fr/cooperation-au-developpement/pays-projets/projets-dans-le-monde/item/150-prd-amelioration-des-systemes-traditionnels-d-elevage-de-petits-ruminants-ovins-et-caprins-dans-un-contexte-de-mutation-environnementale-et-societale-au-benin). The Scientific Committee has approved these protocols. Furthermore, the study involved recording body measurements from sheep with oral consent and in the presence of their owners. Due to the absence of specific legislation for body measurements, no approval was necessary.

### Study area

This study was conducted in the 10 phytogeographic zones (Fig 1) that comprise the three vegetation zones of Benin [22, 23], namely the Guinea-Congolian (GCZ), the Guineo-Sudanian transition (GSZ) zone and the Sudanian zone (SZ). The characteristics of the 10 phytogeographic zones, such as climatic conditions, temperature, humidity index, soil characteristics, and predominant vegetation, are presented in Table 1.

### Sampling procedure

A longitudinal survey was conducted from November 2018 to February 2020 in the 10 phytogeographic zones of Benin. In each zone, two to five communes were selected depending on the presence or abundance of sheep flocks. Thirty-two out of the 77 communes of Benin were included in the survey. At least four distinct villages were randomly chosen from each commune. In each village, 5 to 20 individual flocks were selected based on farmers' willingness to participate in the study. Approximately four or more unrelated animals were sampled per flock based on farmers' knowledge of the individual animals present in their sheep flocks. Thus, a total of 1240 ewes that were at least two years old and multiparous (at least two lambings) were randomly selected, described and phenotypically characterized. The age of animal estimated by the farmers was ascertained by examining their teeth according to the procedure described in [25, 26]. The sampling distribution across the vegetation and phytogeographic

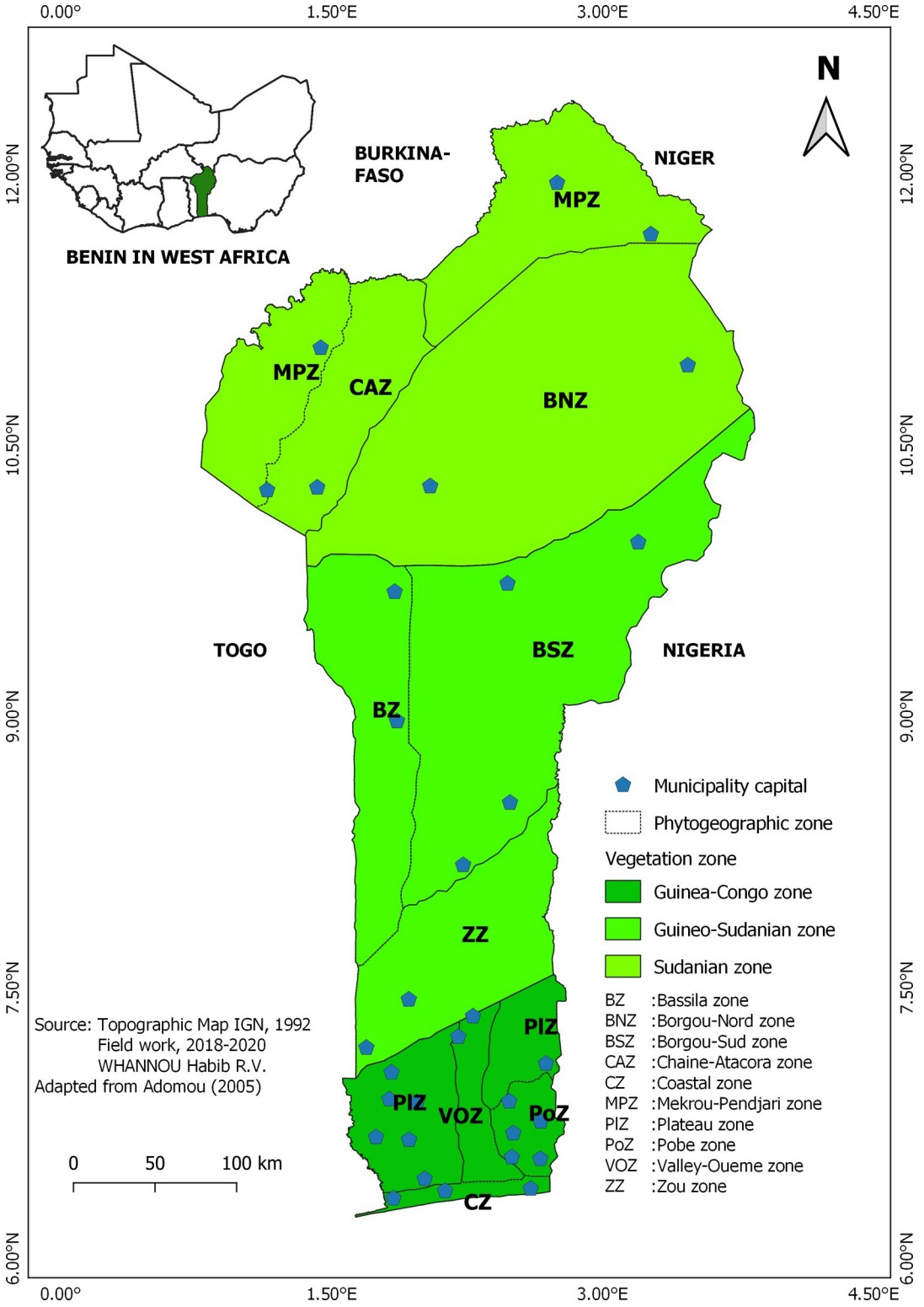

**Fig 1. Map of the vegetation zones and phytogeographic zones of Benin showing the 32 communes sampled to assess the morphological variability within the indigenous sheep population of Benin.** The map was made using QGIS 3.8 [24].

**Table 1. Characteristics of the 10 phytogeographic zones of Benin and sample size (sheep).**

| Vegetation zone | Phyto-geographical zones | Climate | Humidity index | Annual rainfall (mm) | Major soil type | Main vegetation | Sample size |
|---|---|---|---|---|---|---|---|
| GC | Coastal (CZ) | Subequatorial | 4.6 to 5.8 | 900–1300 | Sandy | Dense humid forest, marshy Forest, light forests and savannah | 105 |
| | Pobe (PoZ) | Subequatorial | 4 to 5.8 | 1100–1300 | Ferralitic and without concretions | Dense humid semi-deciduous forest | 96 |
| | Oueme Valley (VOZ) | Subequatorial | 4.9 | 1100–1300 | Hydromorphous with sandy loam to clay loam texture | Marshy forest, forest and sections of dense semi-deciduous forest | 103 |
| | Plateau (PlZ) | Subequatorial | 3.8 to 4.9 | 900–1300 | Red ferralitic soil and without concretions | Dense semi-deciduous forest | 110 |
| GS | Zou (ZZ) | Subhumid tropical | 2.8 | 1100 | Tropical ferruginous | Dense dry forest and light forests | 147 |
| | Bassila (BZ) | Typical subhumid | 2.7 to 3.9 | 1200–1300 | Tropical ferruginous type with ferralitic soil intrusions with concretions | Dense semi-deciduous forest, gallery forests, dense dry forests, Light forests and wooded savannahs | 167 |
| | Borgou-Sud (BSZ) | Tropical with tendency to unimodal | 2.9 | 1200 | Ferruginous soils on crystalline rocks | Light forest and wooded savannahs | 112 |
| S | Borgou-Nord (BNZ) | Typically dry tropical | 1.9 | 1000–1150 | Ferruginous soils on crystalline rocks | Wooded savannahs | 124 |
| | Chaîne Atacora (CAZ) | Typically dry tropical | 2.1 | 1000–1200 | poorly developed, with unrefined minerals | Savannah and Pockets of dense dry Forest and light forests | 148 |
| | Mekrou-Pendjari (MPZ) | Typically dry tropical | 1.9 to 1.4 | 900–1000 | Ferruginous type washed with concretion | Dense dry forests, light forests and wooded savannahs | 128 |
| Total | | | | | | | 1240 |

GC, Guineo-Congolese zone; GS, Guineo-Sudanian zone; S, Sudanian zone.

zones is presented in Table 1. All individuals sampled in a phytogeographic zone were considered as a sub-population.

## Data collection procedure

A total of 25 quantitative linear body measurements (Fig 2 and Table 2) and 5 qualitative physical traits drawn from the FAO guidelines [25] and from a previous study [27], were used to describe the morphological characteristics of each animal. To minimize collecting biases, all measurements were taken by a young researcher and a trained field assistant. The live body-weight of each animal was measured using a scale. The other 24 body measurements were taken using a flexible measuring tape and a measuring stick, early in the morning before the animals were fed to avoid biases on certain traits due to feed intake. In addition, the reproductive history of each sampled animal, including the number and type of parities (single, twins, triplet and quadruplets), was recorded from its owner. The geographical position of the herds in which the sheep individuals were sampled was recorded using a Garmin GPS (etrex vista TM).

## Data analysis

Fourteen morphological indices (Table 3) were calculated based on the collected quantitative linear body measurements (or morphometric traits). All statistical analyses were conducted using SAS version 9.4 (SAS Institute Inc., Cary, NC, USA). Descriptive statistics for the quantitative linear body measurements and qualitative physical traits were obtained using the procedures PROC UNIVARIATE and PROC FREQ, respectively. The frequencies and Pearson chi-

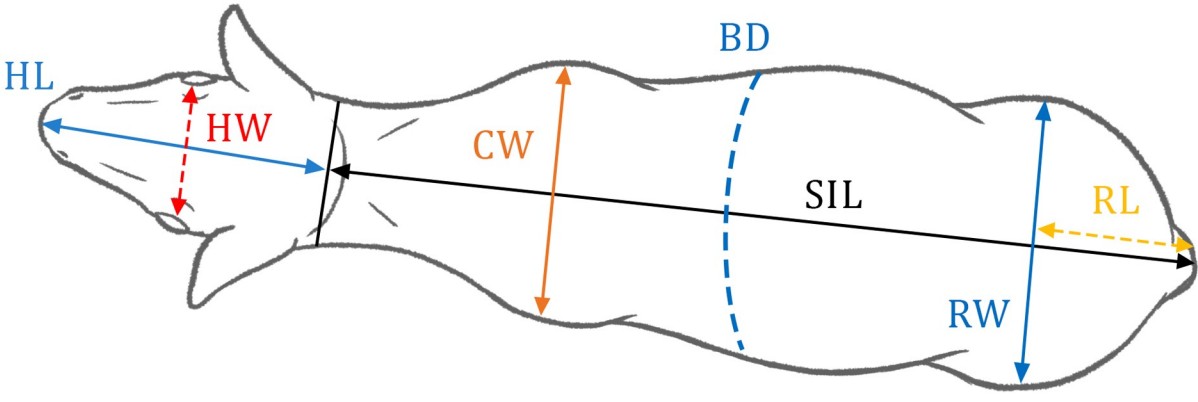

**Fig 2. Illustration of the 25 linear body measurements taken on each sampled sheep.** WH, withers height; RH, rump height; SH, sternum height; BH, back height; CD, chest depth; RD, rump depth; CW, chest width; SIL, scapulo-ischial length; BL, body length; HL, head length; HW, head width; EL, ear length; MD, muzzle diameter; NL, neck length; NG, neck girth; TL, tail length; HG, heart girth; CC, chest circumference; AG, abdominal girth; BD, bicostal diameter; RW, rump width; RL, rump length; TC, cannon bone circumference; HC, hock circumference.; BW, body weight.

**Table 2. Description of linear body measurements (in centimeters, cm) and qualitative traits taken on 1240 adult sheep in Benin.**

| Phenotypic traits | Description |
|---|---|
| **Quantitative linear body measurements** | |
| Withers height (WH) | The (vertical) height from the bottom of the front foot to the highest point of the shoulder between the withers. |
| Rump height (RH) | Distance from the highest point of rump to the ground |
| Sternum height (SH) | Vertical distance between the ground and the sternum, just between the top of the shoulder on its lateral side and the end of the neck. |
| Back height (BH) | Distance from the middle of back to the ground |
| Chest depth (CD) | Vertical distance from the apex of the withers to the bottom of the chest |
| Rump depth (RD) | Vertical distance from the apex of the rump to the bottom of the chest |
| Chest width (CW) | Distance between the extremities of the shoulders just behind the forelegs |
| Scapulo-ischial length (SIL) | The horizontal distance from the head (between the medium of horn sites) to the basis of tail |
| Body length (BL) | Distance from the point of shoulder to the ischial tuberosity |
| Head length (HL) | Distance from the nape to the alveolar edge of the incisors of the upper jaw |
| Head width (HW) | Distance between the external extremities of the eyes |
| Ear length (EL) | Length of the external ear from its root on the poll to the tip |
| Muzzle diameter (MD) | Measure above of the nostril and around of point where whalebone meets the chin |
| Neck length (NL) | Measure from the poll to the center of the withers with the sheep head being held up at approximately 90 degrees to its body |
| Neck girth (NG) | Circumference of the neck in its median part |
| Tail length (TL) | Distance from the tail attachment to the tip of the tail |
| Heart girth (HG) | The circumference of the body immediately behind the shoulder blades in a vertical plane, perpendicular to the long axis of the body |
| Chest circumference (CC) | Circumference of the body at the end of the chest cavity in a vertical plane perpendicular to the long axis of the body |
| Abdominal girth (AG) | Circumference of body passing immediately behind the sacrum and at udder level. |
| Bicostal diameter (BD) | Length of the curve between the ends of the last bones of the chest cage |
| Rump width (RW) | Distance between the two tips of the rump |
| Rump length (RL) | Distance from the middle of the rump tips to the tips of the buttocks just at the base of the tail |
| Cannon bone circumference (TC) | Circumference of the cannon bone at a hand brace below the lower part of the knee joint |
| Hock circumference (HC) | Circumference taken just above the hock joint |
| Body weight (BW) | Fasting weight taken with a scale |
| **Qualitative physical traits** | |
| Facial (head) profile | Straight, Convex, Ultra-convex |
| General aspect of body hair coat color | Plain/Uniform, Pied, Spotted |
| Unique color of coat | Black, White, Brown |
| Other color of coat | Black, White, Brown |
| Back profile | Straight, Slopes up towards the rump, Dipped/Curved |
| Hair type | Long, Short |
| Ear orientation | Erected, Dropped |

square ($\chi2$) tests were used for qualitative physical traits to explore the relationships among qualitative variables. The least-square means (LSmeans), their standard errors (SEs), and the

**Table 3. Morphological indices calculated from quantitative linear body measurements taken on 1240 adult sheep in Benin.**

| Index | Formulas | References |
|---|---|---|
| Mass index (MI) | MI = BW*100/WH | [28] |
| Slenderness index (IGS) | IGS = (WH-CD)/CD | [29–32] |
| Auricular index (IAT) | IAT = EL/CD | [29–32] |
| Sternum index (USI) | USI = (WH-CD)*100/WH | [28] |
| Boniness index (BI) | BI = TC*100/HG | [28] |
| Pelvic (IP) | IP = RW*100/RL | [27] |
| Chest depth index (CDI) | CDI = (CD *100)/WH | [28] |
| Size Index (SI) | SI = WH/BL*100 | [33] |
| Balance (Ba) | Ba = (RL*RW)/(CD*CW) | [27, 34] |
| Body (IBR) | IBR = BL*100/HG | [27, 29, 30] |
| Pectoral Index (PI) | PI = ((SH+RH)/2)/SH | [27] |
| Cephalic (IC) | IC = (HW*100)/HL | [27] |
| Body ratio (BR) | BR = SH/RH | [27] |
| Thoracic development (TD) | TD = HG/SH | [27] |

BW, body weight; WH, withers height; CD, chest depth; EL, ear length; TC, cannon bone circumference; HG, heart girth; RW, rump width; RL, rump length; CW, chest width; BL, body length; SH, sternum height; RH, rump height; HW, head width.

coefficients of variation (CVs) of the morphometric traits were calculated for each phytogeographic zone. The comparison of LSmeans between phytogeographic zones was performed using Tukey's test multiple mean comparison tests. Subsequently, the general linear model procedure (PROC GLM) was used to analyze the relationship between phytogeographic zones and morphometric traits.

A stepwise discriminant analysis was performed using PROC STEPDISC to identify the most useful morphometric traits and morphological indices for further discriminant analyses. The relative discriminatory ability of a quantitative variable was assessed using the partial R-square, F value, and level of significance (Pr>F). Then, the canonical discriminant analysis (CDA) function (PROC CANDISC) was used to perform univariate and multivariate one-way analyses, derive canonical functions and linear combinations of the quantitative variables that summarize variation between populations, and calculate the associated Mahalanobis distances. The ability of the computed canonical functions to classify each individual animal into its *a priori* phytogeographic zone was measured using the discriminant procedure (PROC DISCRIM). The degree of morphological similarity or dissimilarity among individuals from the different phytogeographic zones was determined based on the ascending hierarchical clustering (AHC) analysis procedure (PROC CLUSTER). The PROC TREE procedure was used to build a dendrogram displaying the interrelationships among individuals within and across phytogeographic zones. Finally, a multiple correspondence analysis (MCA) using the PROC CORRESP procedure was used to associate the qualitative physical traits with the phytogeographic zones.

## Results

### Relationships among sheep morphometric traits and the 10 phytogeographic zones of Benin using univariate analyses

The result of the univariate analysis showed significant differences (P<0.05) among the 10 phytogeographic zones for all measured quantitative morphometric variables (S1 Table) and the

calculated morphological indices (S2 Table). Overall, most of the quantitative linear body traits, except ear length (EL), tail length (TL) and body weight (BW), had relatively high CVs, whereas the quantitative variables chest depth (CD), scapulo-ischial length (SIL), body length (BL), muzzle diameter (MD), heart girth (HG), and cannon bone circumference (TC) had relatively low CVs. High values of CVs were observed for most of the quantitative variables in the Borgou-Nord and Chaîne Atacora zones in the North. Most of the morphological indices, except mass index (MI), balance index (Ba), and auricular index (IAT), showed relatively low CVs with the least variation for body ratio (BR), pelvic index (IP), sternum index (USI), body (IBR), size index (SI), chest depth index (CDI), and boniness index (BI). However, for most of the measured morphometric traits and indices, the highest mean values were observed in the Mekrou-Pendjari zone in the North and the lowest mean values in the Oueme Valley and Pobe zones in the South. The highest mean values recorded for withers height (WH), rump height (RH), BL, and SIL were respectively 68.3, 68.8, 63.2 and 91.2 cm, and the lowest were 49.2, 50.9, 50.9 and 69.3 cm respectively. The highest mean values of the slenderness (IGS) and IAT indices were 1.36 and 0.61, respectively, and the lowest mean values were 1.16 and 0.37, respectively (S2 Table).

Significant differences in the frequencies (P< 0.0001) were observed among the 10 phytogeographic zones for certain qualitative traits, such as head profile, coat color and patterns, hair type, back profile, and ear orientation (Table 4). A composite coat color (88.1%) was more frequently observed regardless of the phytogeographic zone, with a dominance of spotted white (75.6%), black (6.6%) and brown (5.9%) colors. It was followed by the plain/uniform white (9.0%) and piebald (2.9%) colors, which appeared to be present almost exclusively in the Mekrou-Pendjari, Borgou-Nord, and Chaîne Atacora phytogeographic zones. Long hair (61.9%) and erected/pendulous ears (72.6%) were more common than short hair (38.1%) and dropped ears (27.4%), which also predominated in the Mekrou-Pendjari, Borgou-Nord, and Chaîne Atacora zones. Additionally, animals with dipped back (44.0%) were relatively more common than those with slopes up towards the rump (31.9%) and straight back (24.1%). Irrespective of the phytogeographic zone, the most commonly observed facial profile was the straight type (60.1%), followed by the convex type (37.5%).

The proportion of birth type varied significantly (p<0.0001) among the 10 phytogeographic zones (S3 Table). Irrespective of the zone, single-born lambs were the most common. The highest percentages of twin-born lambs were recorded in the Oueme Valley, Pobe, and Zou zones, whereas the highest proportions of triplets and quadruplets were recorded in the Pobe zone. The percentage of multiple births appeared to increase with the parity number of ewes.

## Identification of sheep sub-populations using multivariate analyses

The results of the stepwise discriminant analysis (Table 5) showed that 38 out of the 39 quantitative variables (i.e., 25 quantitative linear body traits and 14 morphological indices) included in the analysis significantly contribute to discrimination between the phytogeographic zones (P<0.0001). The traits rump width (RW) and sternum height (SH) showed higher partial $R^2$ and F values, illustrating their greater discriminant power than the other variables used to assess the morphological diversity in the Benin sheep population. Nevertheless, the use of the 32 significant (P<0.0001 for column Pr > F) quantitative variables (i.e., 22 quantitative linear body traits and 10 morphological index) in the CDA generated two significant (P<0.0001) canonical variables (CAN 1 and CAN 2) that explain 76% of the total variation, as revealed by the standardized coefficients for the discriminant function, the canonical correlation, the eigenvalue, and the share of total variance taken into account (Table 6). Canonical loadings that measure the simple linear correlations between each independent variable and canonical variables are reported in Table 6. CAN 1 was dominated by positive loadings of head length

**Table 4. Frequency (%) of qualitative traits in sheep population of the 10 phytogeographic zones of Benin.**

| Zones | BSZ n = 112 | BZ n = 167 | BNZ n = 124 | CAZ n = 148 | MPZ n = 128 | PlZ n = 110 | PoZ n = 96 | VOZ n = 103 | CZ n = 105 | ZZ n = 147 | Overall n = 1240 | Chi$^2$ | P |
|---|---|---|---|---|---|---|---|---|---|---|---|---|---|
| **Facial profile** | | | | | | | | | | | | 498.85 | 0.0001 |
| Convex | 47.3 | 9.0 | 46.8 | 75.7 | 77.3 | 19.1 | 4.2 | 26.2 | 1.0 | 17.7 | 33.6 | | |
| Ultra-convex | 1.8 | 0.0 | 5.6 | 5.4 | 9.4 | 0.0 | 0.0 | 0.0 | 0.0 | 0.7 | 2.4 | | |
| Straight | 50.9 | 91.0 | 47.6 | 18.9 | 13.3 | 80.9 | 95.8 | 73.8 | 99 | 81.6 | 64.0 | | |
| **Coat color** | | | | | | | | | | | | 166.06 | 0.0001 |
| Spotted white | 79.5 | 71.9 | 78.2 | 75.7 | 67.2 | 84.5 | 78.1 | 71.8 | 67.6 | 81.6 | 75.6 | | |
| Spotted brown | 6.3 | 1.8 | 5.6 | 5.4 | 7.8 | 5.5 | 5.2 | 11.7 | 6.7 | 5.4 | 5.9 | | |
| Spotted black | 7.1 | 5.4 | 4.0 | 6.1 | 3.9 | 3.6 | 11.5 | 8.7 | 19.0 | 1.4 | 6.6 | | |
| Zoned pie | 2.7 | 1.2 | 4.8 | 3.4 | 14.8 | 0.0 | 0.0 | 0.0 | 0.0 | 0.7 | 2.9 | | |
| Plain/Uniform white | 4.5 | 19.8 | 7.3 | 9.5 | 6.3 | 6.4 | 5.2 | 7.8 | 6.7 | 10.9 | 9.0 | | |
| **Hair length** | | | | | | | | | | | | 395.67 | 0.0001 |
| Long | 43.8 | 76.6 | 16.9 | 33.8 | 28.1 | 81.8 | 91.7 | 83.5 | 93.3 | 83.0 | 61.9 | | |
| Short | 56.3 | 23.4 | 83.1 | 66.2 | 71.9 | 18.2 | 8.3 | 16.5 | 6.7 | 17.0 | 38.1 | | |
| **Back profile** | | | | | | | | | | | | 1004.33 | 0.0001 |
| Straight | 13.4 | 4.2 | 31.5 | 36.5 | 10.9 | 45.5 | 8.3 | 52.4 | 1.9 | 38.1 | 24.1 | | |
| Curved/Dipped | 78.6 | 5.4 | 62.1 | 62.8 | 85.9 | 50.0 | 1.0 | 20.4 | 4.8 | 58.5 | 44.0 | | |
| Slopes up towards the rump | 8.0 | 90.4 | 6.5 | 0.7 | 3.1 | 4.5 | 90.6 | 27.2 | 93.3 | 3.4 | 31.9 | | |
| **Ear orientation** | | | | | | | | | | | | 392.51 | 0.0001 |
| Erected/Pendulous | 76.8 | 77.2 | 54.8 | 47.3 | 17.2 | 91.8 | 94.8 | 94.2 | 96.2 | 91.8 | 72.6 | | |
| Dropped | 23.2 | 22.8 | 45.2 | 52.7 | 82.8 | 8.2 | 5.2 | 5.8 | 3.8 | 8.2 | 27.4 | | |

MPZ, Mekrou-Pendjari zone; CAZ, Chaîne Atacora zone; BNZ, Borgou-Nord zone; BSZ, Borgou-Sud zone; BZ, Bassila zone; CZ, Coastal zone; PoZ, Pobe zone; PlZ, Plateau zone; VOZ, Oueme Valley zone; ZZ, Zou zone.

P is the probability observed for the qualitative traits; Chi$^2$ expresses independence between variables at 5% level.

(HL), SH, BR, SI, neck length (NL), MD, IAT, EL, negative loadings of cephalic index (IC), thoracic development (TD), Ba, RW and IP. In contrast, CAN 2 was dominated by positive loadings of BL, TL, back height (BH), RH, WH, CD, BW, HG, TC, rump depth (RD), head width (HW), MI and neck length (NL).

The plot of the centroid values of the first two canonical discriminant functions (CAN1 and CAN2) showed many distinct and homogenous sheep sub-populations with overlapping events (Fig 3).

The Mahalanobis distances among the 10 phytogeographic zones are presented in Table 7. All pairwise distances were significant (P<0.0001). The two largest measured squared Mahalanobis distances were between the Mekrou-Pendjari and Pobe zones (69.02) and between the Oueme Valley and Bassila zones (61.65). The closest distance (2.85) was between the Chaîne Atacora and Borgou-Nord zones. The discriminant functions accurately classified a relatively high proportion (74.59%) of the individual sheep into their *a priori* group (Table 8).

Based on the squared Mahalanobis distances, AHC generated a dendrogram that indicated four distinct sub-groups or sub-populations of sheep (Fig 4). The first sub-population was composed of Borgou-Nord and Chaîne Atacora zones joined by the Borgou-Sud zone, the second was only composed of the Mekrou-Pendjari zone, the third was composed of the Pobe and Costal zones joined by the Bassila zone, and the fourth was composed of the Oueme Valley, Zou, and Plateau zones.

**Table 5. Summary of the stepwise selection of quantitative traits obtained from the stepwise discriminant analysis performed on 39 morphometric variables.**

| Step | Number of traits | Partial R2 | F value | Pr > F | Wilks' lambda (λ) | Pr < Lambda (λ) | Average squared canonical correlation | Pr >ASCC |
|------|------------------|-----------|---------|--------|-------------------|-----------------|--------------------------------------|----------|
| 1 | RW | 0.7377 | 384.46 | < .0001 | 0.26225 | < .0001 | 0.082 | < .0001 |
| 2 | SH | 0.6708 | 278.31 | < .0001 | 0.08632 | < .0001 | 0.143 | < .0001 |
| 3 | HC | 0.3351 | 68.76 | < .0001 | 0.05740 | < .0001 | 0.175 | < .0001 |
| 4 | HL | 0.3240 | 65.36 | < .0001 | 0.03880 | < .0001 | 0.203 | < .0001 |
| 5 | TL | 0.2567 | 47.05 | < .0001 | 0.02884 | < .0001 | 0.222 | < .0001 |
| 6 | Ba | 0.2359 | 42.02 | < .0001 | 0.02203 | < .0001 | 0.243 | < .0001 |
| 7 | SI | 0.1744 | 28.73 | < .0001 | 0.01819 | < .0001 | 0.259 | < .0001 |
| 8 | CC | 0.1615 | 26.16 | < .0001 | 0.01525 | < .0001 | 0.273 | < .0001 |
| 9 | IAT | 0.1620 | 26.24 | < .0001 | 0.01278 | < .0001 | 0.288 | < .0001 |
| 10 | TD | 0.1560 | 25.08 | < .0001 | 0.01079 | < .0001 | 0.301 | < .0001 |
| 11 | BH | 0.1294 | 20.14 | < .0001 | 0.00939 | < .0001 | 0.307 | < .0001 |
| 12 | HG | 0.1112 | 16.94 | < .0001 | 0.00835 | < .0001 | 0.315 | < .0001 |
| 13 | EL | 0.1146 | 17.53 | < .0001 | 0.00739 | < .0001 | 0.325 | < .0001 |
| 14 | MI | 0.1132 | 17.26 | < .0001 | 0.00655 | < .0001 | 0.331 | < .0001 |
| 15 | CW | 0.1290 | 20.01 | < .0001 | 0.00571 | < .0001 | 0.338 | < .0001 |
| 16 | IC | 0.1054 | 15.91 | < .0001 | 0.00511 | < .0001 | 0.347 | < .0001 |
| 17 | CM | 0.0873 | 12.89 | < .0001 | 0.00466 | < .0001 | 0.352 | < .0001 |
| 18 | IP | 0.0823 | 12.08 | < .0001 | 0.00428 | < .0001 | 0.358 | < .0001 |
| 19 | CD | 0.1190 | 18.19 | < .0001 | 0.00377 | < .0001 | 0.366 | < .0001 |
| 20 | BW | 0.0754 | 10.97 | < .0001 | 0.00349 | < .0001 | 0.370 | < .0001 |
| 21 | WH | 0.1452 | 22.84 | < .0001 | 0.00298 | < .0001 | 0.377 | < .0001 |
| 22 | TC | 0.0784 | 11.42 | < .0001 | 0.00275 | < .0001 | 0.381 | < .0001 |
| 23 | NG | 0.0642 | 9.22 | < .0001 | 0.00257 | < .0001 | 0.385 | < .0001 |
| 24 | NL | 0.0497 | 7.01 | < .0001 | 0.00244 | < .0001 | 0.390 | < .0001 |
| 25 | BD | 0.0475 | 6.68 | < .0001 | 0.00233 | < .0001 | 0.393 | < .0001 |
| 26 | HW | 0.0387 | 5.39 | < .0001 | 0.00224 | < .0001 | 0.397 | < .0001 |
| 27 | RD | 0.0372 | 5.18 | < .0001 | 0.00215 | < .0001 | 0.399 | < .0001 |
| 28 | USI | 0.0321 | 4.43 | < .0001 | 0.00208 | < .0001 | 0.400 | < .0001 |
| 29 | IBR | 0.0309 | 4.25 | < .0001 | 0.00202 | < .0001 | 0.403 | < .0001 |
| 30 | BL | 0.0307 | 4.23 | < .0001 | 0.00196 | < .0001 | 0.406 | < .0001 |
| 31 | SIL | 0.0273 | 3.75 | 0.0001 | 0.00190 | < .0001 | 0.408 | < .0001 |
| 32 | AG | 0.0261 | 3.58 | 0.0002 | 0.00185 | < .0001 | 0.410 | < .0001 |
| 33 | IGS | 0.0238 | 3.24 | 0.0007 | 0.00181 | < .0001 | 0.412 | < .0001 |
| 34 | BI | 0.0220 | 3.00 | 0.0015 | 0.00177 | < .0001 | 0.413 | < .0001 |
| 35 | PI | 0.0155 | 2.09 | 0.0276 | 0.00174 | < .0001 | 0.414 | < .0001 |
| 36 | RH | 0.0363 | 5.00 | < .0001 | 0.00168 | < .0001 | 0.416 | < .0001 |
| 37 | BR | 0.0284 | 3.88 | < .0001 | 0.00163 | < .0001 | 0.418 | < .0001 |
| 38 | RL | 0.0147 | 1.98 | 0.0387 | 0.00161 | < .0001 | 0.418 | < .0001 |

Number of traits is the number of variables in the model; F, F value for entering or removing the variable; Pr> F, the probability level for the Fstatistic; Pr<Lambda is based on the F approximation to Wilks' lambda; Pr>ASCC is based on the F approximation to Pillai's trace.

## Multiple correspondence analysis of sheep qualitative traits

Multiple correspondence analysis (MCA) highlighted the association between the different qualitative physical traits and phytogeographic zones (Fig 5). The first two dimensions (Dim 1 and Dim 2) explained 68.88% and 18.21% of the total variation, respectively. On the right-hand side of the plot, the Zou, Plateau and Oueme Valley zones were closely associated with

**Table 6. Total canonical coefficients for the canonical function, the adjusted canonical correlation, the eigenvalue, the approximate standard error of the canonical correlations and the percentage total variance accounted for obtained from the canonical discriminant analysis performed on 32 morphometric variables.**

| Variables | Can1 | Can2 |
|---|---|---|
| WH | 0.43 | 0.64 |
| RH | 0.30 | 0.65 |
| SH | 0.72 | 0.41 |
| BH | 0.30 | 0.65 |
| CD | 0.26 | 0.60 |
| RD | 0.26 | 0.55 |
| CW | -0.27 | 0.27 |
| BL | 0.07 | 0.72 |
| HL | 0.74 | 0.41 |
| HW | 0.03 | 0.55 |
| EL | 0.49 | 0.47 |
| CM | 0.53 | 0.25 |
| NL | 0.57 | 0.50 |
| NG | 0.14 | 0.46 |
| TL | 0.28 | 0.69 |
| HG | 0.13 | 0.58 |
| CC | 0.02 | 0.38 |
| BD | 0.39 | 0.47 |
| RW | -0.80 | 0.45 |
| TC | 0.22 | 0.57 |
| HC | -0.18 | -0.07 |
| BW | 0.25 | 0.59 |
| MI | 0.08 | 0.52 |
| IAT | 0.52 | 0.30 |
| IS | 0.42 | 0.28 |
| IP | -0.73 | 0.34 |
| SI | 0.68 | 0.28 |
| Ba | -0.80 | 0.18 |
| IBR | -0.24 | 0.00 |
| IC | -0.83 | 0.02 |
| BR | 0.70 | 0.13 |
| TD | -0.82 | -0.08 |
| Adjusted Canonical Correlation | 0.942 | 0.871 |
| Approximate Standard Error of the canonical correlations | 0.003 | 0.007 |
| Eigenvalue | 8.0952 | 3.2562 |
| Proportion of the eigenvalue sum | 0.54 | 0.08 |
| Cumulative proportion of the eigenvalue sum | 0.54 | 0.84 |

sheep with a straight back, long hair, a predominant coat color of spotted white and brown patterns. Animals from the Pobe, Coastal, and Bassila zones were characterized by back profile slopes up towards the rump and either a plain/uniform white or a composite coat color with predominantly spotted black or white patterns. Moreover, the left-hand side shows that the Mekrou-Pendjari zone was plainly associated with sheep that had dropping ears, flush hairs, an ultra-convex head profile, and pie-black or pie-brown coat color. Conversely, the zones of Borgou-Sud, Borgou-Nord and Chaîne Atacora were associated with sheep that have a convex

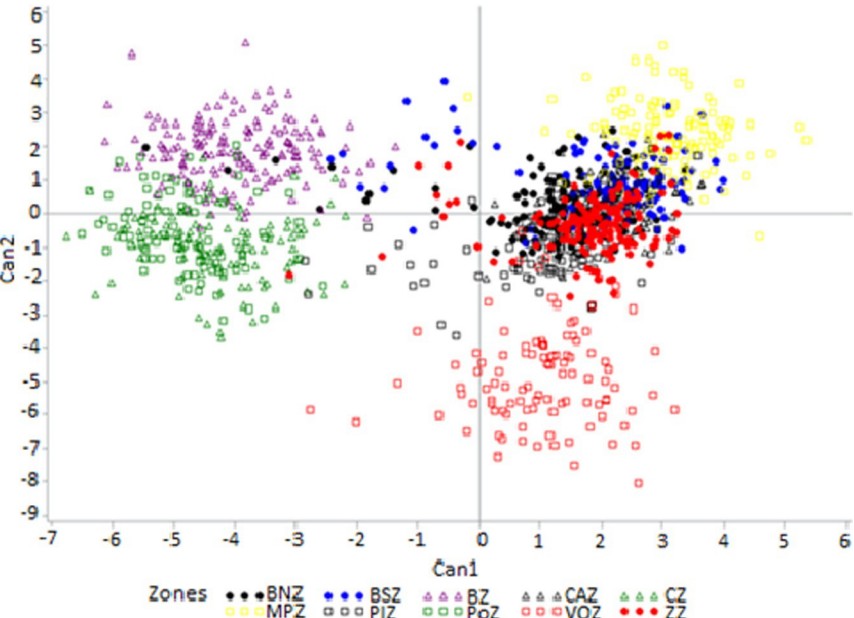

**Fig 3. Scatterplot visualizing the 1240 individual sheep sampled from the ten phytogeographic zones of Benin on the first two canonical discriminant functions.** Colored symbols group correspond to phytogeographic zones. MPZ, Mekrou-Pendjari zone; CAZ, Chaîne Atacora zone; BNZ, Borgou-Nord zone; BSZ, Borgou-Sud zone; BZ, Bassila zone; CZ, Coastal zone; PoZ, Pobe zone; PlZ, Plateau zone; VOZ, Oueme Valley zone; ZZ, Zou zone.

head profile, a dipped back profile and a coat color with predominantly spotted white and brown patterns (Fig 5).

## Discussion

In this study, we aimed to further document the existing diversity and spatial distribution within the sheep population raised in Benin based on a large panel of qualitative and quantitative traits. Univariate analyses revealed significant differences among phytogeographic zones for all measured morphological traits and derived indices, suggesting the possible influence of these zones on the evolutionary adaptation of the sheep population in terms of these

**Table 7. Mahalanobis distances between sheep populations identified by phytogeographic zones (n = 1240) obtained from the canonical discriminant analysis.**

| Zones | BNZ | BSZ | BZ | CAZ | CZ | MPZ | PlZ | PoZ | VOZ | ZZ |
|-------|-----|-----|-----|-----|-----|-----|-----|-----|-----|-----|
| BNZ | 00.00 | | | | | | | | | |
| BSZ | 09.24 | 00.00 | | | | | | | | |
| BZ | 34.79 | 40.72 | 00.00 | | | | | | | |
| CAZ | 02.85 | 05.12 | 43.63 | 00.00 | | | | | | |
| CZ | 33.88 | 44.77 | 15.96 | 41.80 | 00.00 | | | | | |
| MPZ | 18.51 | 11.52 | 60.98 | 12.38 | 61.06 | 00.00 | | | | |
| PlZ | 04.93 | 10.19 | 39.22 | 04.59 | 33.97 | 22.84 | 00.00 | | | |
| PoZ | 41.62 | 50.13 | 14.03 | 49.29 | 04.38 | 69.02 | 39.26 | 00.00 | | |
| VOZ | 29.88 | 27.32 | 61.65 | 25.20 | 49.76 | 45.96 | 20.52 | 56.58 | 00.00 | |
| ZZ | 08.40 | 05.44 | 44.30 | 06.25 | 40.31 | 19.34 | 06.07 | 46.85 | 21.21 | 00.00 |

MPZ, Mekrou-Pendjari zone; CAZ, Chaîne Atacora zone; BNZ, Borgou-Nord zone; BSZ, Borgou-Sud zone; BZ, Bassila zone; CZ, Coastal zone; PoZ, Pobe zone; PlZ, Plateau zone; VOZ, Oueme Valley zone; ZZ, Zou zone.

**Table 8. Percentage of individual sheep classified into phytogeographic zones (n = 1240) based on discriminant analysis.**

| Zones | Posterior probability (%) | | | | | | | | | |
|---|---|---|---|---|---|---|---|---|---|---|
| | **BNZ** | **BSZ** | **BZ** | **CAZ** | **CZ** | **MPZ** | **PlZ** | **PoZ** | **VOZ** | **ZZ** |
| BNZ | **64.52** | 00.00 | 02.42 | 13.71 | 01.61 | 02.42 | 12.90 | 00.81 | 00.00 | 01.61 |
| BSZ | 04.46 | **68.75** | 05.36 | 07.14 | 00.89 | 03.57 | 00.89 | 00.00 | 00.89 | 08.04 |
| BZ | 00.00 | 00.00 | **88.62** | 00.00 | 03.59 | 00.00 | 00.00 | 07.78 | 00.00 | 00.00 |
| CAZ | 10.14 | 08.11 | 00.00 | **55.41** | 00.00 | 04.73 | 08.78 | 00.00 | 00.00 | 12.84 |
| CZ | 00.00 | 00.00 | 03.81 | 00.00 | **70.48** | 00.00 | 00.00 | 25.71 | 00.00 | 00.00 |
| MPZ | 00.00 | 07.03 | 00.00 | 07.81 | 00.00 | **83.59** | 00.78 | 00.00 | 00.00 | 00.78 |
| PlZ | 08.18 | 02.73 | 00.91 | 05.45 | 01.82 | 00.91 | **68.18** | 00.00 | 01.82 | 10.00 |
| PoZ | 00.00 | 00.00 | 07.29 | 00.00 | 14.58 | 00.00 | 00.00 | **78.13** | 00.00 | 00.00 |
| VOZ | 00.00 | 00.00 | 00.00 | 01.94 | 00.00 | 00.00 | 04.85 | 00.97 | **89.32** | 02.91 |
| ZZ | 10.36 | 08.84 | 00.68 | 02.72 | 00.68 | 02.72 | 04.08 | 00.00 | 00.00 | **78.91** |
| Rate | 00.36 | 00.31 | 00.11 | 00.45 | 00.30 | 00.16 | 00.32 | 00.22 | 00.11 | 00.21 |
| Priors | 00.10 | 00.10 | 00.10 | 00.10 | 00.10 | 00.10 | 00.10 | 00.10 | 00.10 | 00.10 |

MPZ, Mekrou-Pendjari zone; CAZ, Chaîne Atacora zone; BNZ, Borgou-Nord zone; BSZ, Borgou-Sud zone; BZ, Bassila zone; CZ, Coastal zone; PoZ, Pobe zone; PlZ, Plateau zone; VOZ, Oueme Valley zone; ZZ, Zou zone.

Values above and/or below the diagonal represent the percentage of individuals from other phytogeographic zones present in the zone considered by the diagonal value.

Rate: proportion of misclassified observation in each phytogeographic zone.

Priors: Prior probabilities of group membership.

morphological traits. This result is in line with the finding of a previous study [16], who reported a significant impact of the breeding area on morphological traits in the sheep population from Ivory Coast. The mean values of thoracic development (TD), greater than 1.2,

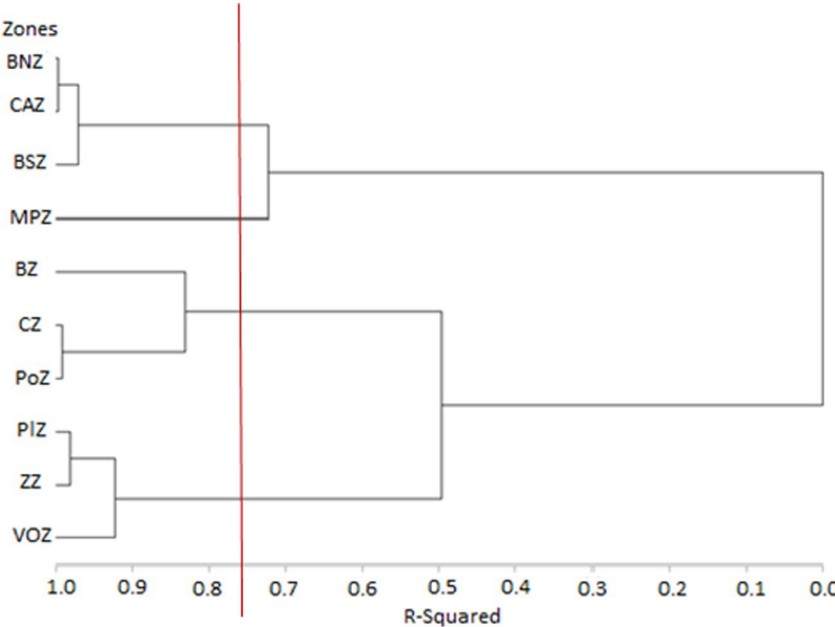

**Fig 4. Clusterdness of sheep population in Benin based on the Mahalanobis distance computed by ascending hierarchical cluster analysis (AHC).** MPZ, Mekrou-Pendjari zone; CAZ, Chaîne Atacora zone; BNZ, Borgou-Nord zone; BSZ, Borgou-Sud zone; BZ, Bassila zone; CZ, Coastal zone; PoZ, Pobe zone; PlZ, Plateau zone; VOZ, Oueme Valley zone; ZZ, Zou zone.

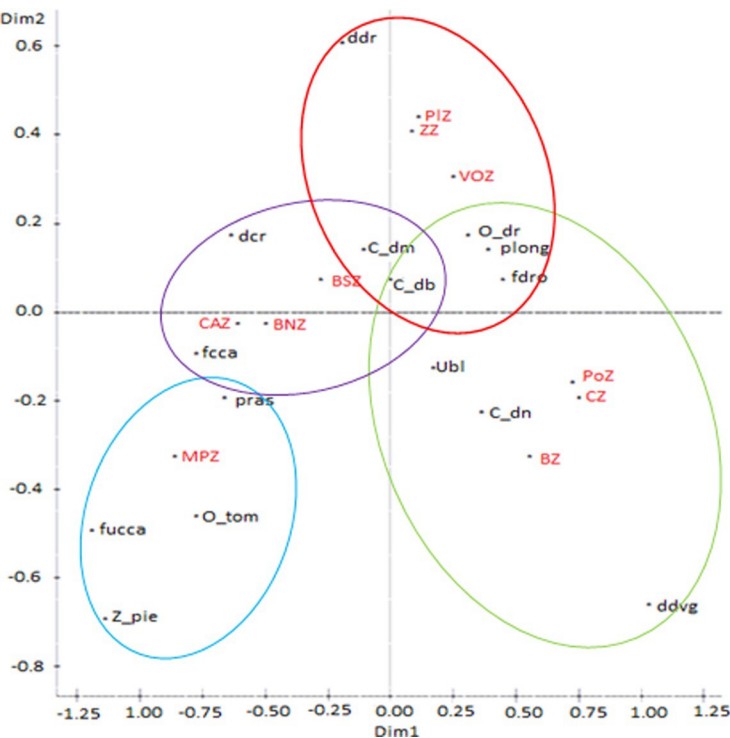

**Fig 5. Multiple correspondence analysis of the qualitative physical traits of the sheep population of Benin.** MPZ, Mekrou-Pendjari zone; CAZ, Chaîne Atacora zone; BNZ, Borgou-Nord zone; BSZ, Borgou-Sud zone; BZ, Bassila zone; CZ, Coastal zone; PoZ, Pobe zone; PlZ, Plateau zone; VOZ, Oueme Valley zone; ZZ, Zou zone; ddr, Straight back; dcr, Dipped/curved back; ddvg, Back slopes up towards the rump; Z_pie, Pied-bald; fcca, Convex facial; fucca, Ultra-convex facial; fdro, Straight facial; O_tom, Dropped ear; O_dr, Erected ear; Ubl, Plain/Uniform white; C_dn, Spotted black; C_db, Spotted white; C_dm, dominant brown; plong, Long hair; pras, Short hair.

indicated that all measured animals had good thoracic development, regardless of the phytogeographic zone. However, following the classification based on the body index (IBR) [27], the sheep from the phytogeographic zones of Borgou-Nord, Chaîne Atacora, Plateau, and to some extent Borgou-Sud, can be considered as of the brevigline breed (mean values of IBR < 0.85), whereas those of Bassila, Mekrou-Pendjari, Pobe, Oueme Valley, Coastal and Zou zones are of the medigline breed (mean values of IBR > 0.85). The overall calculated cephalic index (IC) of 51.20% indicated that sheep from Benin are dolichocephalic, regardless of the phytogeographic zone. This result is considerably lower than that reported in a previous study [35] for sheep breeds in Nigeria. Based on the main qualitative traits generally used for breed description (i.e., coat color, facial profile, ear orientation), the composite coat color with a dominance of spotted white associated with both straight facial profile, long hairs, and erected ears, most frequently recorded in the Southern zones (i.e., BZ, PlZ, PoZ, CZ, ZZ and VOZ) are physical characteristics of Djallonké/WAD sheep, as described in previous studies [10, 16, 36]. This result suggests that the sheep in the aforementioned zones are closer to the Djallonké sheep. In contrast, specific physical traits of the long-legged Sahelian sheep breed, such as convex facial profile, short hair, and dropped ears orientation, were more common in the sub-populations surveyed in the Northern zones (i.e., BSZ, BNZ, CAZ, and MPZ). Some individuals from the Sahelian sheep breed presented a bicolored coat (the front being brown or black and the rear white), mainly in Mekrou-Pendjari (MPZ). These characteristics are specific to Fulani sheep, also known as Oudah. Notably, in these Northern zones, some of the long-legged Sahelian sheep breed's physical characteristics were also observed in crossbreeding products between Sahelian and West

African Dwarf sheep. This distribution of sheep populations along the South-North gradient was confirmed by the results of the multiple correspondence analysis for qualitative physical traits. This finding could be explained either by the increasing introgression of Sahelian long-legged sheep from the Sahel through transhumance and trade or by the selection pressure for specific traits. The mobility of herders with diverse sheep breeds in West Africa could favor genetic introgression and be a dynamic factor of animal genetic diversity. According to a recent report on sheep transhumance between Niger and Benin [37], transhumant sheep herders move from Niger towards localities of the Alibori department in North Benin and stay for about six months to ensure feeding and watering of their animals. Therefore, many exchanges are made between transhumant and local breeders favoring the mixture of Sahelian or long-legged type sheep (i.e., Oudah, Bali-Bali, and Balami) with WAD sheep. The distribution gradient could also result from an adaptive response to changing local environments since morphological adaptations (body size and shape, coat and skin color, and hair type) are physical changes in the animal that enhance its fitness in a given environment [38, 39].

The multivariate discriminant analyses confirmed the significant morphological variability among sheep from most of the 10 phytogeographic zones of Benin. The proximity between sheep from CZ, PoZ, and BZ and their farness from those of VOZ, MPZ, BNZ, BSZ, CAZ, ZZ, and PlZ (Fig 3) could be explained by the geographic proximity and exchange of animals, as well as by similarities or dissimilarities in their ecologies [7]. Nonetheless, the low proportion of individuals from CAZ (55.4%) and BNZ (64.5%) correctly classified in their origin groups, as well as the low values of Mahalanobis distance between these two zones, revealed some overlap among sheep breeds of these phytogeographic zones with those from other zones except for BZ, PoZ, CZ, and VOZ. These overlaps could be the result of crossbreeding, especially in the BNZ and CAZ phytogeographic zones that host transhumant sheep flocks. Additionally, the subdivision of the sheep population into four sub-populations in the hierarchical cluster analysis (Fig 4) seems to reflect differences in the type of vegetation, climate, and humidity among phytogeographic zones. This result is likely to confirm the effect of environmental factors on the morphology of sheep [40–43] and transhumance and management practices. Many convergences with other reports on the measured morphological traits were found in the current study. For example, the mean values of the two quantitative body linear traits, WH and HG, obtained for the sheep from the Southern zones (i.e., PlZ, ZZ and VOZ), (53.2±3.40 cm and 60.9±1.67 cm, respectively), were highly similar to previously reported values (52.3 ± 1.07 cm and 65.0 ± 1.60 cm, respectively) for the Djallonké/WAD sheep in Nigeria [44], and to the WH mean value of 56.5± 0.22 cm reported for the Djallonké/WAD sheep in Burkina Faso [31]. Therefore, the sheep sub-population found in these zones appeared to be an ecotype of the Djallonké sheep (WAD). Likewise, the mean value of these traits (HW and HG) obtained for the sheep from PoZ, CZ and BZ (52.9±5.07 cm and 73.6±3.72 cm, respectively) were highly similar to those (54.6 ± 8.23 cm and 74.7 ± 8.28 cm, respectively) reported for the Djallonké/WAD sheep in Togo [36]. Thus, the sheep from PoZ, CZ, and BZ might represent another ecotype of Djallonké/WAD sheep with a size relatively larger than the first sheep sub-group of PlZ, ZZ, and VOZ. As for the sheep population from MPZ, which is dominated by individuals with physical characteristics of the Sahelian long-legged sheep, the mean value of WH obtained in the current study (i.e., 68.4±0.47 cm) was similar to that (69.1 ±0.12 cm) reported for the Sahelian sheep in Burkina Faso [31]. The mean values of WH and HG for the sheep from BNZ, CAZ, and BSZ (61.2±0.92 cm and 68.9±0.90 cm, respectively) were intermediate between those for the sheep from MPZ and each of the two other groups of Southern zones (PlZ, ZZ, and VOZ; PoZ, CZ, and BZ) (Figs 3 and 4), suggesting that these zones may be considered as very favorable zones for crossbreeding. This result also suggests the co-existence of several sheep morphotypes in these zones.

This study highlights a highly diverse sheep population in Benin, as in other African countries (e.g., Burkina Faso, Ivory Coast, Togo, and Nigeria), within which the distribution of individuals is affected by natural and also anthropogenic factors. Thus, the sheep subgroups observed in the different phytogeographic zones of Benin also exist in other African countries in similar or different environments [16, 31, 36, 44]. The most important natural factors at the origin of the recorded sheep diversity across the 10 investigated phytogeographic zones might be climate-related factors (temperature, humidity, and/or vegetation cover), which affect the availability of feed resources and induce natural selection pressures. Anthropogenic factors mainly concern animal management practices in different zones, cultural preferences, and livestock marketing systems. Thus, the phenotypic traits (small size, stocky appearance, small ears, and long hair) of the Djallonké/WAD sheep, which are predominantly found in Southern Benin, are likely a response to natural selection over several generations under the influence of the constraints of the environment in which the animals are raised. Furthermore, the larger phenotypic traits of the Djallonké ecotype in the PoZ, CZ, and BZ could be explained, in addition to the influence of the environment, by changes in sheep farmers' breeding practices in these phytogeographic zones, especially the practice of crossbreeding short-legged with long-legged animals from the North. This is undoubtedly influenced by the annual flow of Sahelian animals to these regions during the Aid El-Kebir cultural ceremonies when sheep sacrifice takes place in Muslim households. In addition, the breeders of these areas would try to adapt to new consumer demands, as expressed by their preference for animals that possess larger physical features than the Djallonké during ceremonies and festivals. Likewise, the Sahelian sheep, which are predominant in the MPZ in northern Benin, are larger and slender, with varied but predominantly light coats, short hair, a long tail, dropped, and larger ears. Several hypotheses about the adaptive value of these traits have been put forth. For instance, [45] argue that these specific traits might allow them to reflect solar radiation better, and thus, to be less prone to heat stress. In addition, according to these authors their long legs might predispose them to travel long distances when searching for pastures. Moreover, their large height might allow them to feed easily in tree and shrubs savannah pastures, which are predominant in these regions [22, 23]. But confirmation of these hypotheses requires further study and remains inconclusive. The BNZ and CAZ, with their intermediate climatic gradient between the humid south and the dry north, promote, on the one hand, the extension of the distribution area of the Sahelian types, and on the other hand, the cross-border sheep transhumance practices that are at the origin of the admixture of sub-populations observed in these two zones. Referring to transhumance, it is worth noting that during the migratory period, and to meet their own subsistence needs, transhumant sheepherders often sell or exchange a few heads of animals in their herds for food and salt [37]. In contrast, the attraction of certain sheep farmers for large animals in areas hosting transhumant-herds sometimes encourages them to herd their animals to the same grazing areas in the hope of mating their animals with those kept by the transhumant herders.

Although morphological variation is largely under genetic control [30], it is subject to the influence of the environment and management practices [46, 47]. Thus, the preservation of local populations that adapt to their environment is essential. This calls for the development of new management strategies for sheep farming in Benin as well as in other African countries aiming to improve farm profitability by improving animal performance while preserving the diversity within the local sheep populations. In this way, sheep farming would overcome current and future challenges in production systems in Africa, including climate change and market demand.

## Conclusion

This study aimed to explore the morphological variability of indigenous sheep reared in different phytogeographic zones of Benin. The results showed significant variations in phenotypic

traits, both qualitative and quantitative, among phytogeographic zones. Four sheep sub-populations were identified. Animals in the phytogeographic zones of Southern Benin could be identified as short-legged (Djallonké/WAD) sheep, whereas those from the zones located in the northern regions of the country were much closer to the long-legged Sahelian sheep breed. The intermediate sub-populations included an ecotype of Djallonké/WAD sheep and various crossbreeds. These results could be due to several factors, such as adaptation of animals or natural selection, changes in farmers' breeding practices, and gene flow. Further research is ongoing to better understand the genetic, environmental, and socio-economic determinants of these recorded morphological variations. Thereafter, a breeding program could be developed and implemented for better management of the diversity existing between and within recorded sheep sub-populations and for the sustainable production of this livestock species in Benin.

## Supporting information

**S1 Fig. Djallonké ewe and lambs.**
(TIFF)

**S2 Fig. Djallonké ram.**
(TIFF)

**S3 Fig. Sahelian ewe.**
(TIF)

**S4 Fig. Sahelian ram.**
(TIFF)

**S5 Fig. Crossbreed ewe.**
(TIFF)

**S6 Fig. Crossbreed ram.**
(TIFF)

**S1 Table. Least squares means (LSmeans), standard errors (SEs) and coefficients of variation (CVs) of morphological measurements (cm) across phytogeographic zones.**
(PDF)

**S2 Table. Least squares means (LSmeans) standard errors (SEs) and coefficients of variation (CVs) of morphological indices across phytogeographic zones.**
(PDF)

**S3 Table. Incidence of phytogeographic zones and number of parity on the type of parity.**
(PDF)

**S1 Database.**
(XLSX)

## Acknowledgments

The authors would like to acknowledge sheep farmers from the different phytogeographic zones of Benin for their consent and active participation in this study. We also thank Daphné Braun for her contribution to the description of the morphometric measurements used in this study.

## Author Contributions

**Conceptualization:** Habib Rainier Vihotogbe Whannou, Martin Spanoghe, Deborah Lanterbecq, Dominique Demblon, Marcel Romuald Benjamin Houinato, Luc Hippolyte Dossa.

**Data curation:** Habib Rainier Vihotogbe Whannou, Cossi Ulriche Afatondji, Luc Hippolyte Dossa.

**Formal analysis:** Habib Rainier Vihotogbe Whannou, Cossi Ulriche Afatondji, Maurice Cossi Ahozonlin, Luc Hippolyte Dossa.

**Funding acquisition:** Dominique Demblon, Marcel Romuald Benjamin Houinato, Luc Hippolyte Dossa.

**Investigation:** Habib Rainier Vihotogbe Whannou, Cossi Ulriche Afatondji.

**Methodology:** Habib Rainier Vihotogbe Whannou, Cossi Ulriche Afatondji, Maurice Cossi Ahozonlin, Luc Hippolyte Dossa.

**Project administration:** Dominique Demblon, Marcel Romuald Benjamin Houinato.

**Supervision:** Luc Hippolyte Dossa.

**Writing – original draft:** Habib Rainier Vihotogbe Whannou, Maurice Cossi Ahozonlin, Luc Hippolyte Dossa.

**Writing – review & editing:** Habib Rainier Vihotogbe Whannou, Martin Spanoghe, Deborah Lanterbecq, Dominique Demblon, Marcel Romuald Benjamin Houinato, Luc Hippolyte Dossa.

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
