## [Decision Letter · Decision Letter 0]

5 May 2021

PONE-D-21-06521

Morphological variability within the indigenous sheep population reared in Benin (West Africa)

PLOS ONE

Dear Dr. DOSSA,

Thank you for submitting your manuscript to PLOS ONE. After careful consideration, we feel that it has merit but does not fully meet PLOS ONE’s publication criteria as it currently stands. Therefore, we invite you to submit a revised version of the manuscript that addresses the points raised during the review process.

The manuscript represents an interesting study of a noteworthy subject.

The reviewers have mainly concerns about structure and writing, but not repeat analyses or experiments.

Partial list of their recommendations, add background information on breeds in Benin and nearby countries, better scientific focus (goat diversity, adaptive changes, breeding focus), grammatical editing, description of trait values and changes in them, check the CPA analyses and revisit the conclusions.

We look forward to receiving your revised manuscript.

Kind regards,

Arnar Palsson, Ph.D.

Academic Editor

PLOS ONE

Additional Editor Comments:

The manuscript represents an interesting study of a noteworthy subject.

The reviewers have mainly concerns about structure and writing, but not repeat analyses or experiments.

Partial list of their recommendations, add background information on breeds in Benin and nearby countries, better scientific focus (goat diversity, adaptive changes, breeding focus), grammatical editing, description of trait values and changes in them, check the CPA analyses and revisit the conclusions.

I would also say it that you could rework the title. (skipping "reared" and "West Africa" I think), perhaps??

"Morphological variability within the indigenous sheep population(s)?? of Benin"

Journal Requirements:

2. We note that Figure 1 in your submission contain map images which may be copyrighted. All PLOS content is published under the Creative Commons Attribution License (CC BY 4.0), which means that the manuscript, images, and Supporting Information files will be freely available online, and any third party is permitted to access, download, copy, distribute, and use these materials in any way, even commercially, with proper attribution. For these reasons, we cannot publish previously copyrighted maps or satellite images created using proprietary data, such as Google software (Google Maps, Street View, and Earth). For more information, see our copyright guidelines: http://journals.plos.org/plosone/s/licenses-and-copyright.

2.1.    You may seek permission from the original copyright holder of Figure 1 to publish the content specifically under the CC BY 4.0 license. 

2.2.    If you are unable to obtain permission from the original copyright holder to publish these figures under the CC BY 4.0 license or if the copyright holder’s requirements are incompatible with the CC BY 4.0 license, please either i) remove the figure or ii) supply a replacement figure that complies with the CC BY 4.0 license. Please check copyright information on all replacement figures and update the figure caption with source information. If applicable, please specify in the figure caption text when a figure is similar but not identical to the original image and is therefore for illustrative purposes only.

3. Please include captions for *all* your Supporting Information files at the end of your manuscript, and update any in-text citations to match accordingly. Please see our Supporting Information guidelines for more information: http://journals.plos.org/plosone/s/supporting-information.

Reviewers' comments:

Reviewer's Responses to Questions

**Comments to the Author**

1. Is the manuscript technically sound, and do the data support the conclusions?

Reviewer #1: Partly

Reviewer #2: Yes

2. Has the statistical analysis been performed appropriately and rigorously? 

Reviewer #1: Yes

Reviewer #2: Yes

3. Have the authors made all data underlying the findings in their manuscript fully available?

Reviewer #1: Yes

Reviewer #2: Yes

4. Is the manuscript presented in an intelligible fashion and written in standard English?

Reviewer #1: Yes

Reviewer #2: Yes

5. Review Comments to the Author

Reviewer #1: This work has a great of interest , it aims to study the morphological variability within the sheep population in Benin as a 19 prelude for their molecular characterization.

This work needs a major revision :

- The introduction need to be more large, we need to get more information about the importance of this adaptive population, the importance of the study of molecular characterization to futur selection scheme

- You should have more information about the animal material, the management, the performances

- In the result part, i twill be better to add the CPA analysis to see more the variability according to the fixed effects used

- The conclusion should be revised again

- we needs a grammatical revision to all the paper

Reviewer #2: PONE-D-21-06521

General comments

In this manuscript, body measurements, indices based on those, and qualitative traits such as color and ear type were studied for 1240 adult ewes from different areas in Benin. Four types/subpopulations of animals were detected and similarities to sheep in nearby countries were discussed. The study is motivated by the need for knowledge about genetic diversity. In the abstract, it is mentioned that it is a prelude to molecular (genetic?) characterization, but this is not mentioned again in the rest of the manuscript. It would be good to clarify if that is the intention for future studies.

The manuscript is generally well written, and it is important to characterize the local genetic diversity of indigenous domestic animal populations, which seems not to be done previously for sheep in Benin. The large number of traits and areas (and abbreviations!) mentioned makes it very important to be clear and consistent in the description and this could be improved to some degree, especially in the tables that need a bit more explanations.

To a foreign reader, it is not very clear if there are any defined sheep breeds in the country at all, for example “exotic” breeds, in addition to the indigenous? What type of production are the sheep used for? Mainly meat production or also e.g. milk and does this differ between the areas studied?

The discussion could be made a bit more interesting. In the discussion there is some mentioning of different sheep types in different areas due to evolutionary adaptions, but this is not well covered. Are there e.g. different climatic zones that make different ear or fur color types more beneficial for survival, or are such trait differences mainly a result of what humans favored for other reasons? The same with e.g. body size or type, is there for example a logical connection to smaller body size in areas with harsher conditions and less feed/grass supply, or has it more to do with what the sheep have been used/selected for (such as meat production) etc.?

Specific comments

Line 57: rewrite so that it is easier to understand which reference you mean by According to [9] without searching in the ref. list. (e.g. According to FAO [9]…)

Table 1: Does the à mean to? Perhaps a – would be clearer.

Line 91-95: Was pedigree records available to select unrelated animals, or how did you determine which animals were unrelated? It appears as if no records were kept about date of birth as an inspection of teeth was made to determine the age, so were there then records about parents?

Line 100: Rewrite so that you do not just say from [14], but make it easier to read and understand, e.g. by saying from a previous study…

Table 2: Some descriptions of conformation measures are unclear, for example the meaning of head medium in the description of SIL, and extremities of eyes (HW), and measure a few above… for MD (a few what?), and in NL from beginning of the throat at its middle? Please go through the table and clarify, and one or several pictures illustrating the different measures would be very helpful.

Line 122: Should it be ..the highest discriminating…?

Line 159: It would be good to remind the reader about the meaning of the IGS index as well, as was done for the other mentioned indices in this part of the text.

Line 162: Do you here mean significant differences in frequencies?

Line 174: Do you mean the ..most common..?

Line 178: Frequency …..traits in sheep populations…?

Table 4: An explanation of the zone abbreviations is needed, and so is an explanation of what the chi and P-values are for. The color names Dominant white etc. can be a bit confusing as they are not the same as used in the text, and are often used in other articles to describe certain genes or inheritance patterns of colors. The n= in the table seem to be the same for all traits in each zone, and should then not be repeated multiple times, it could be put in the first or second row. The back profile is a bit unclear, in the text it says slopes up towards the withers (higher at withers than at rump?) and in Table 4 it says descending towards withers which sounds like the opposite?

Line 185: is the PR>F needed?

Table 5: an explanation of what is meant by Number in traits should be given, and the table description could be a bit longer and more informative. Consider if all given decimals are needed, e.g. for the average squared canonical correlations?

Table 6: Also for this table it would be helpful with a bit more explanations, are the SE for the correlations, proportion of.., cumulative what?

Table 8: Again, some more clear descriptions could be provided: What is rate for example – proportion of animals from one zone classified in another/wrong zone?

Line 250-251: How do you know these are due to evolutionary adaptation? Could not differences in size be due to different management and feeding intensity in addition to genetic adaptation? And there may have been selection for growth and for colors that the animal owners in certain area prefer?

Line 251: the trait thoracic depth has not been defined previously, only thoracic development.

Line 292-293: What do you mean by crossbreeding in this case, are there defined breeds that are crossed or do you mean that animals from different regions or of different types are crossbred?

Line 297-299: do you really define crossbreeding (and natural selection) as an environmental factor (as opposed to genetic)?

6. PLOS authors have the option to publish the peer review history of their article (what does this mean?). If published, this will include your full peer review and any attached files.

Reviewer #1: No

Reviewer #2: No

---

## [Author Response · Author response to Decision Letter 0]

8 Jul 2021

Responses to comments

Editor Comments

Comment: I would also say it that you could rework the title. (skipping "reared"and "West Africa" I think), perhaps??

Response: Title has been changed as follows: "Morphological variability within the indigenous sheep population of Benin)"

Comment: additional requirements

Response

1. Care has been taken to ensure that our manuscript meets PLOS ONE's style requirements.

2. Fig 1: Another map has been produced and the authors have been quoted as recommended.

Reviewer #1 comments

Comment: The introduction need to be larger, we need to get more information about the importance of this adaptive population, the importance of the study of molecular characterization to future selection scheme

Response: The introduction has been revised accordingly, as follows: 

Lines 53-93: “It is commonly accepted that two main types of sheep, the Djallonké (S1 and S2 Figs) and the Sahelian (S3 and S4 Figs), are encountered throughout the country [10,11,12]. Djallonké, also named West African Dwarf sheep, seemed to originate from the type of fine-tailed and hairy sheep native to Western Asia, having migrated to Africa through the Isthmus of Suez and Bab el Mandeb, and was the only type of sheep type on the continent until the third millennium BC. [13]. Widely distributed in West Africa, the Djallonké sheep is mainly raised for meat [10,11,14]. It is particularly adapted to coastal areas [15] because of its resistance to trypanosomiasis [12,16]. However, it would have undergone significant phenotypic changes over time [9,12]. Generally, Djallonké sheep are assimilated to small-sized animals with straight facial profile, small narrow erected ears and a hairy short coat [14,19]. Contrarily to ewes, rams are horned and have a heavy white or pied mane with black forequarters and white hindquarters. Existence in this sheep type of two sub-types differentiated by size has been distinguished [18,19]: the larger in the Sudanian zone and the smaller in the Guinean zone further south [9,17]. 

Sahelian sheep regroup all long legged sheep breeds known under different ethnical and local names in the semi-arid and arid zones of West-African Sahel [9]. Like Djallonké sheep, Sahelian sheep are thought to be descended from the fine-tailed and hairy sheep [9]. Although notorious for not surviving in humid areas [20], they are increasingly encountered in different humid localities of Benin for recent years [11,12], reflecting their progressive adaptation to less dry climates. In the Sahelian pastoral and agro-pastoral production systems, they are used for meat and milk production [14]. Sahelian sheep have concave facial profile, pendulous long ears, a long thin tail and diverse coat colour [9]. A typical characteristic of Sahelian ram is the absence of mane. As in several West African countries, many crossbreeds between Sahelian and Djallonké sheep (S5 and S6 Figs) are present in Benin with various intermediate body sizes.

The lack of consistent knowledge on the genetic diversity of West African sheep populations, as well as on their specific traits constitutes the major constraint for the implementation of sound programs towards their genetic improvement and sustainable use. Moreover, the presence of unknown sub-groups within each of these two known groups of sheep, as well as the occurrence of crossbreeding can lead to certain ambiguities when it comes to distinguish certain individuals according to well-defined breed standards. According to FAO (2012) [21], phenotypic and molecular characterizations are important tools for breed documentation, a first step towards the development of strategies for their sustainable use, management and conservation. To date, neither of these two characterization tools have not been covered in depth for the Beninese sheep population. Hence, in order to further document the existing diversity and to explore the spatial distribution within the indigenous sheep population of Benin, the morphological characterization based on a large panel of collected morphology/phenotypic traits was considered here as a primary study. We hypothesized that the sheep population of Benin is greatly diverse and unevenly distributed according to ecological conditions.

The current study aims at establishing the relationships among sheep morphometric traits and the ten phytogeographical zones of Benin using univariate analyses and then, explore the presence of sheep subpopulations in the Beninese indigenous sheep population using multivariate analyses. The findings of this study will provide the basis for a sound molecular study on the same samples, based on both Simple Sequence Repeat (SSR) and Single Nucleotide Polymorphic (SNP) markers genotyping. Morphological data could then be compared to molecular data and association analyses (i.e. Genome-wide association studies) performed to appropriately address possible breeding strategies for the indigenous sheep population of Benin.”

Comment: You should have more information about the animal material, the management, the performances

Response: More information about these recommendations was included in the introduction, as indicated above.

Comment: In the result part, it will be better to add the CPA analysis to see more the variability according to the fixed effects used

Response: The Canonical discriminant analysis (CANDISC) is a dimension reduction technique which included CPA analysis. This multivariate statistical technique helps to identify differences among groups of individuals and improve understanding the relationships among the variables measured within those groups. More information on this procedure could be funded in “Cruz-Castillo et al 1994.

Comment: The conclusion should be revised again

Response: The conclusion has been revised, as follows: 

Lines 418-430: “The aim of this study was to explore the morphological variability of indigenous sheep reared in different phytogeographical zones of Benin. The results showed significant variations in phenotypic traits, both qualitative and quantitative among phytogeographical zones. Four sheep sub-populations were distinguished. Animals in the phytogeographical zones of Southern Benin could be assimilated to short-legged or WAD sheep, whereas those from the zones located in the northern regions of the country were much closer to the long-legged Sahelian sheep type. The intermediate sub-populations appeared to include an ecotype of WAD sheep and various crossbreeds. These results could be due to several factors like animal’s adaptation or natural selection, changes in farmers’ breeding practices, and gene flow. However, it appears that the indigenous sheep population of Benin has been subjected to very little selective breeding. Further research is ongoing to better understand the genetic, environmental and socio-economic determinants of the recorded morphological variation. A breeding program could therefore be developed and implemented for a better management of the diversity existing between and within recorded sheep sub-populations and for a sustainable production of this livestock species in Benin.”

Comment: we needs a grammatical revision to all the paper

Response: All the paper has been grammatically revised

 

Reviewer #2 comments

General comments

Comment: In this manuscript, body measurements, indices based on those, and qualitative traits such as color and ear type were studied for 1240 adult ewes from different areas in Benin. Four types/subpopulations of animals were detected and similarities to sheep in nearby countries were discussed. The study is motivated by the need for knowledge about genetic diversity. In the abstract, it is mentioned that it is a prelude to molecular (genetic?) characterization, but this is not mentioned again in the rest of the manuscript. It would be good to clarify if that is the intention for future studies.

Response: This has been clarified in the introduction, as follows: 

L 78-93: “According to FAO (2012) [21], phenotypic and molecular characterizations are important tools for breed documentation, a first step towards the development of strategies for their sustainable use, management and conservation. To date, neither of these two characterization tools have not been covered in depth for the Beninese sheep population. Hence, in order to further document the existing diversity and to explore the spatial distribution within the indigenous sheep population of Benin, the morphological characterization based on a large panel of collected morphology/phenotypic traits was considered here as a primary study. We hypothesized that the sheep population of Benin is greatly diverse and unevenly distributed according to ecological conditions.

The current study aims at establishing the relationships among sheep morphometric traits and the ten phytogeographical zones of Benin using univariate analyses and then, explore the presence of sheep subpopulations in the Beninese indigenous sheep population using multivariate analyses. The findings of this study will provide the basis for a sound molecular study on the same samples, based on both Simple Sequence Repeat (SSR) and Single Nucleotide Polymorphic (SNP) markers genotyping. Morphological data could then be compared to molecular data and association analyses (i.e. Genome-wide association studies) performed to appropriately address possible breeding strategies for the indigenous sheep population of Benin.”

Comment: The manuscript is generally well written, and it is important to characterize the local genetic diversity of indigenous domestic animal populations, which seems not to be done previously for sheep in Benin.

The large number of traits and areas (and abbreviations!) mentioned makes it very important to be clear and consistent in the description and this could be improved to some degree, especially in the tables that need a bit more explanations.

Response: This has been improved through the clarification of some trait definitions in the Table 2. Moreover, a graphical illustration of all the morphometric measurements used in the study is now suggested as a new figure (FIG 2).

Comment: To a foreign reader, it is not very clear if there are any defined sheep breeds in the country at all, for example "exotic" breeds, in addition to the indigenous? What type of production are the sheep used for? Mainly meat production or also e.g. milk and does this differ between the areas studied?

The discussion could be made a bit more interesting. In the discussion there is some mentioning of different sheep types in different areas due to evolutionary adaptions, but this is not well covered. Are there e.g. different climatic zones that make different ear or fur color types more beneficial for survival, or are such trait differences mainly a result of what humans favored for other reasons? The same with e.g. body size or type, is there for example a logical connection to smaller body size in areas with harsher conditions and less feed/grass supply, or has it more to do with what the sheep have been used/selected for (such as meat production) etc.?

Information about breeds/types have been added as follows:

Lines 53-93: “It is commonly accepted that two main types of sheep, the Djallonké (S1 and S2 Figs) and the Sahelian (S3 and S4 Figs), are encountered throughout the country [10,11,12]. Djallonké, also named West African Dwarf sheep, seemed to originate from the type of fine-tailed and hairy sheep native to Western Asia, having migrated to Africa through the Isthmus of Suez and Bab el Mandeb, and was the only type of sheep type on the continent until the third millennium BC. [13]. Widely distributed in West Africa, the Djallonké sheep is mainly raised for meat [10,11,14]. It is particularly adapted to coastal areas [15] because of its resistance to trypanosomiasis [12,16]. However, it would have undergone significant phenotypic changes over time [9,12]. Generally, Djallonké sheep are assimilated to small-sized animals with straight facial profile, small narrow erected ears and a hairy short coat [14,19]. Contrarily to ewes, rams are horned and have a heavy white or pied mane with black forequarters and white hindquarters. Existence in this sheep type of two sub-types differentiated by size has been distinguished [18,19]: the larger in the Sudanian zone and the smaller in the Guinean zone further south [9,17]. 

Sahelian sheep regroup all long legged sheep breeds known under different ethnical and local names in the semi-arid and arid zones of West-African Sahel [9]. Like Djallonké sheep, Sahelian sheep are thought to be descended from the fine-tailed and hairy sheep [9]. Although notorious for not surviving in humid areas [20], they are increasingly encountered in different humid localities of Benin for recent years [11,12], reflecting their progressive adaptation to less dry climates. In the Sahelian pastoral and agro-pastoral production systems, they are used for meat and milk production [14]. Sahelian sheep have concave facial profile, pendulous long ears, a long thin tail and diverse coat colour [9]. A typical characteristic of Sahelian ram is the absence of mane. As in several West African countries, many crossbreeds between Sahelian and Djallonké sheep (S5 and S6 Figs) are present in Benin with various intermediate body sizes.

The lack of consistent knowledge on the genetic diversity of West African sheep populations, as well as on their specific traits constitutes the major constraint for the implementation of sound programs towards their genetic improvement and sustainable use. Moreover, the presence of unknown sub-groups within each of these two known groups of sheep, as well as the occurrence of crossbreeding can lead to certain ambiguities when it comes to distinguish certain individuals according to well-defined breed standards. According to FAO (2012) [21], phenotypic and molecular characterizations are important tools for breed documentation, a first step towards the development of strategies for their sustainable use, management and conservation. To date, neither of these two characterization tools have not been covered in depth for the Beninese sheep population. Hence, in order to further document the existing diversity and to explore the spatial distribution within the indigenous sheep population of Benin, the morphological characterization based on a large panel of collected morphology/phenotypic traits was considered here as a primary study. We hypothesized that the sheep population of Benin is greatly diverse and unevenly distributed according to ecological conditions.

The current study aims at establishing the relationships among sheep morphometric traits and the ten phytogeographical zones of Benin using univariate analyses and then, explore the presence of sheep subpopulations in the Beninese indigenous sheep population using multivariate analyses. The findings of this study will provide the basis for a sound molecular study on the same samples, based on both Simple Sequence Repeat (SSR) and Single Nucleotide Polymorphic (SNP) markers genotyping. Morphological data could then be compared to molecular data and association analyses (i.e. Genome-wide association studies) performed to appropriately address possible breeding strategies for the indigenous sheep population of Benin.”

Discussion 

Lines 383-416: “This study highlights a highly diverse Beninese sheep population, within which the distribution of individuals is affected by natural and also anthropic factors. The most important natural factors at the origin of the recorded sheep diversity across the ten investigated phytogeographical zones are climate-related (temperature, humidity and/or vegetation cover) which affect the availability of feed resources and induce natural selection pressures. The anthropic factors mainly concern animal management practices in the different zones, cultural preferences and livestock marketing systems. Thus, the phenotypic traits (small size, stocky appearance, small ears, long hair), characteristics of the Djallonké sheep type that predominates in Southern Benin are likely a response to natural selection over several generations under the influence of the constraints of the environment in which the animals are raised. Furthermore, the larger phenotypic traits of the Djallonké ecotype in the PoZ, CZ and BZ could be explained, in addition to the influence of the environment, by changes in sheep farmers’ breeding practices in these phytogeographical zones, in particular the practice of crossbreeding short-legged with Long legged animals from the North. This is undoubtedly influenced by the annual flow of Sahelian animals to these regions during the cultural ceremonies of Aid El-Kebir, when sheep sacrifice takes place in Muslim households. In addition, the breeders of these areas would try to adapt to new consumer demands, as expressed by their preference for animals that possess larger physical features than the Djallonké during ceremonies and festivals. Likewise, the Sahelian sheep, which is predominant in the MPZ in North Benin, are larger and slenderer, with varied but predominantly light coats, a short hair, a long tail, dropped and larger ears. These specific traits allow them to withstand heat stress and to adapt to the savannah vegetation of trees and shrubs, predominant in these areas [44], that would be relatively harsher for small-sized sheep. Furthermore, their long legs predispose them to travel long distances in search of pastures. The BNZ and CAZ, with their intermediate climatic gradient between the humid South and the dry North, promote, on the one hand the extension of the distribution area of the Sahelian types, and on the other hand the cross-border sheep transhumance practices that are at the origin of the admixture of subpopulations observed in these two zones. Referring to transhumance, it is worth noting that during the migratory period, and in order to meet their own subsistence needs, transhumant sheep herders often sell or exchange a few heads of animals in their herds for food and salt [36]. On the other hand, the attraction of certain sheep farmers for large animals in areas hosting transhumant-herds sometimes encourages them to herd their animals to the same grazing areas in the hope of mating their animals with those kept by the transhumant herders.

Although morphological variation is largely under genetic control [29], it is subject to the influence of the environment and management practices. Thus, the preservation of local populations that adapt to their environment is essential. This calls for the development of new management strategies for sheep farming in Benin aiming at improving farm profitability through improved animal performances while preserving the diversity within the local sheep population in order to face current and future challenges in production systems, including climate change, and respond to market demand.

Specific comments

Comment: Line 57: rewrite so that it is easier to understand which reference you mean by According to [9] without searching in the ref. list. (e.g. According to FAO [9] …).

Response: this sentence has been rewritten

 Line 78-79: “According to FAO (2012) [21], phenotypic and molecular characterizations……”

Comment: Table 1: Does the à mean to? Perhaps a - would be clearer.

Response: The term “the” has been added

Lines 110-111: “… predominant vegetation are presented in the Table 1.”

Comment: Line 91-95: Was pedigree records available to select unrelated animals, or how did you determine which animals were unrelated? It appears as if no records were kept about date of birth as an inspection of teeth was made to determine the age, so were there then records about parents?

Response: No, with a few exceptions there was no written data on the pedigree of the animals for most of the farms. (No formal birth registration) but the breeders knew their animals well, the period of birth, the origin... The sampling was therefore done on the basis of their memory, the information provided by the breeders on the pedigree of the animals and/or their origin. This information has been now added and the sentence rephrased as follows: About 4 or more unrelated animals were sampled per flock based on farmers’ knowledge of the individual animals present in their sheep flocks.

Comment: Line 100: Rewrite so that you do not just say from [14], but make it easier to read and understand, e.g. by saying from a previous study…

Response: This sentence has been rewritten

Line 133: “… traits drawn from the FAO guidelines [24] and from a previous study [26],...”

Comment: Table 2: Some descriptions of conformation measures are unclear, for example the meaning of head medium in the description of SIL, and extremities of eyes (HW), and measure a few above… for MD (a few what?), and in NL from beginning of the throat at its middle? Please go through the table and clarify, and one or several pictures illustrating the different measures would be very helpful.

Response: 

- The definitions of SIL, BL, HW, MD and NL have been reviewed. 

- new figure has been added to better describe the different morphometric measurements taken (Fig 2).

Comment: Line 122: Should it be ..the highest discriminating…?

Response: This sentence has been rewritten as follows:

(Line 162-163): A stepwise discriminant analysis was performed using the PROC STEPDISC to identify the most useful morphometric traits and morphological indices for further discriminant analyses

Comment: Line 159: It would be good to remind the reader about the meaning of the IGS index as well, as was done for the other mentioned indices in this part of the text. 

Response: This sentence has been rewritten as follows:

Line 201: The highest mean values of the slenderness (IGS)

Comment: Line 162: Do you here mean significant differences in frequencies?

Response: Yes. This sentence has been rewritten

Line 203:” Significant differences in frequencies…”

Comment: Line 174: Do you mean the ..most common..?

Response: Yes. This sentence has been rewritten

Line 216: “Irrespective of the zone, single birth-kids were the most common.”

Comment: Line 178: Frequency...traits in sheep populations…?

Response: Yes. The tittle of the Table 4 has been rewritten

Line 221: “Table 4. Frequency (%) of qualitative traits in sheep population…”

Comment: Table 4: An explanation of the zone abbreviations is needed, and so is an explanation of what the chi and P-values are for. The color names Dominant white etc. can be a bit confusing as they are not the same as used in the text, and are often used in other articles to describe certain genes or inheritance patterns of colors. The n= in the table seem to be the same for all traits in each zone, and should then not be repeated multiple times, it could be put in the first or second row. The back profile is a bit unclear, in the text it says slopes up towards the withers (higher at withers than at rump?) and in Table 4 it says descending towards withers which sounds like the opposite?

Response: The zone abbreviations were described, and chi2 and P-values were explained below the Table 4. The qualitative variable denomination, such as the color names, have been reviewed in the Table 4 as well as in the manuscript. The n= has rewritten in the first lines and repetition has been deleted in the Table 4. The definition of back profile modalities has been corrected in the Table 4 and uniformed as well in the manuscript.

*(No, the height at withers is smaller than the height at rump)

Comment: Line 185: is the PR>F needed?

Response: Yes, it allows to specify the level of choice for the thirty-two significant variables included in the rest of the analyzes.

Comment: Table 5: an explanation of what is meant by Number in traits should be given (the number of variables in the model), and the table description could be a bit longer and more informative. Consider if all given decimals are needed, e.g. for the average squared canonical correlations? 

Response: Explanation of what is meant by Number in traits has been given below the table 5.

The table description has been improved in the Table title and further described in the manuscript: Lines 229-233: “The traits RW and SH showed higher partial R2 and F-values illustrating their greater discriminant power compared with the other variables used to assess the morphological diversity in Benin sheep population. Nevertheless, the use of the thirty-two significant (P<0.0001 for column Pr > F) quantitative variables (i.e., 22 quantitative linear body traits and 10 morphological index) in the”. 

The decimal numbers have been rounded for Wilks’ lambda (λ) and average squared canonical correlations added in the Table 5.

Comment: Table 6: Also for this table it would be helpful with a bit more explanations, are the SE for the correlations, proportion of.., cumulative what?

Response: Yes it is the approximate SE of the canonical correlations. More information has been given for these in the Table 6, as follows:

Approximate Standard Error = Approximate Standard Error of the canonical correlations, Proportion = Proportion of the eigenvalue sum

cumulative = Cumulative proportion of the eigenvalue sum

Comment: Table 8: Again, some more clear descriptions could be provided: What is rate for example - proportion of animals from one zone classified in another/wrong zone?

Response: The description of Rate and Priors has been given below the Table 8 as follows:

Rate: proportion of misclassified observation in each phytogeographical zone

Priors: Priors probabilities of group membership

Comment: Line 250-251: How do you know these are due to evolutionary adaptation? Could not differences in size be due to different management and feeding intensity in addition to genetic adaptation? And there may have been selection for growth and for colors that the animal owners in certain area prefer?

Response: 

- We have just suspected that the variations are an adaptive response. This sentence has been rewritten in Lines 312-318: “In this study, we aimed to further document the existing diversity and the spatial distribution within the sheep population raised in Benin based on a large panel of collected qualitative and quantitative traits. Univariate analyses revealed significant differences among phytogeographical zones for all measured morphological traits and derived indices, suggesting possible influence of these zones on the evolutionary adaption of the sheep population regarding these morphological traits. This result is in line with findings of N’Goran et al. (2019) [16] who reported a significant impact of the breeding area on morphological traits in the sheep population from Ivory-Coast.”

- Yes, differences in size could also be due to different management in addition to genetic adaptation and this has been added in the discussion, as follows:

Lines 382-410: “This study highlights a highly diverse Beninese sheep population, within which the distribution of individuals is affected by natural and also anthropic factors. The most important natural factors at the origin of the recorded sheep diversity across the ten investigated phytogeographical zones are climate-related (temperature, humidity and/or vegetation cover) which affect the availability of feed resources and induce natural selection pressures. The anthropic factors mainly concern animal management practices in the different zones, cultural preferences and livestock marketing systems. Thus, the phenotypic traits (small size, stocky appearance, small ears, long hair), characteristics of the Djallonké sheep type that predominates in Southern Benin are likely a response to natural selection over several generations under the influence of the constraints of the environment in which the animals are raised. Furthermore, the larger phenotypic traits of the Djallonké ecotype in the PoZ, CZ and BZ could be explained, in addition to the influence of the environment, by changes in sheep farmers’ breeding practices in these phytogeographical zones, in particular the practice of crossbreeding short-legged with Long legged animals from the North. This is undoubtedly influenced by the annual flow of Sahelian animals to these regions during the cultural ceremonies of Aid El-Kebir, when sheep sacrifice takes place in Muslim households. In addition, the breeders of these areas would try to adapt to new consumer demands, as expressed by their preference for animals that possess larger physical features than the Djallonké during ceremonies and festivals. Likewise, the Sahelian sheep, which is predominant in the MPZ in North Benin, are larger and slenderer, with varied but predominantly light coats, a short hair, a long tail, dropped and larger ears. These specific traits allow them to withstand heat stress and to adapt to the savannah vegetation of trees and shrubs, predominant in these areas [44], that would be relatively harsher for small-sized sheep. Furthermore, their long legs predispose them to travel long distances in search of pastures. The BNZ and CAZ, with their intermediate climatic gradient between the humid South and the dry North, promote, on the one hand the extension of the distribution area of the Sahelian types, and on the other hand the cross-border sheep transhumance practices that are at the origin of the admixture of subpopulations observed in these two zones. Referring to transhumance, it is worth noting that during the migratory period, and in order to meet their own subsistence needs, transhumant sheep herders often sell or exchange a few heads of animals in their herds for food and salt [36]. On the other hand, the attraction of certain sheep farmers for large animals in areas hosting transhumant-herds sometimes encourages them to herd their animals to the same grazing areas in the hope of mating their animals with those kept by the transhumant herders.”

Comment: Line 251: the trait thoracic depth has not been defined previously, only thoracic development.

Response: This definition was an error. The right definition has been corrected in Line 318: “The mean values of thoracic development (TD), greater…”

Comment: Line 292-293: What do you mean by crossbreeding in this case, are there defined breeds that are crossed or do you mean that animals from different regions or of different types are crossbred?

Response: We mean that in these areas farmers cross different types from the same location and from different regions. Now, this sentence has been deleted to avoid repetition in the text.

Comment: Line 297-299: do you really define crossbreeding (and natural selection) as an environmental factor (as opposed to genetic)?

Response: No, it is an error of formulation. This sentence has been improved as follows: Lines 362-363: “This result thus confirms the influence of environmental factors on the morphology of sheep [39-42], in addition to transhumance and management practices.”

---

## [Decision Letter · Decision Letter 1]

26 Jul 2021

PONE-D-21-06521R1

Morphological variability within the indigenous sheep population of Benin

PLOS ONE

Dear Dr. DOSSA,

Thank you for submitting your manuscript to PLOS ONE. After careful consideration, we feel that it has merit but does not fully meet PLOS ONE’s publication criteria as it currently stands. Therefore, we invite you to submit a revised version of the manuscript that addresses the points raised during the review process.

We look forward to receiving your revised manuscript.

Kind regards,

Arnar Palsson, Ph.D.

Academic Editor

PLOS ONE

Journal Requirements:

Additional Editor Comments (if provided):

PLOS_BeninSheep

The manuscript is greatly improved. There are however several issues remaining.

1. We asked for the results to be put into a broader focus. The background you provided is all about sheep in Benin, but this can be scaled back and a broader geographic view taken. This makes the study more general, to students of sheep on the African continent and elsewhere. What is know about radiation or adaptive evolution of sheep breeds in other countries or in other parts of Africa?

2. Improve the grammar and wording. Get an outsider to review the paper for you.

Examples of sentences that need improving

Abstract: “Good knowledge…”

Line 50 “In West Africa, sheep populations are raised under harsh and diverse ecological conditions, which may have led to the evolution of…”

Line 65 “Djallonké sheep are assimilated to small-sized”?? better verb?

Line 79 drop “consistent”

Line 85

“To date, neither of these two characterization tools have not been covered in depth for of the Beninese sheep populations.”????

There are several examples where extra words are used, that can be dropped. E.g. 510 “fully acknowledge”

3. Indicate the origin of the map, software, database etc.

4. Figure legends should be extended. They should better describe the content of the figure.

5. Tone downs statements of adaptive value of traits throughout manuscript.

a. For instance in discussion (line 441) “These specific traits may allow them to withstand heat stress and to adapt”

Reviewers' comments:

Reviewer's Responses to Questions

**Comments to the Author**

1. If the authors have adequately addressed your comments raised in a previous round of review and you feel that this manuscript is now acceptable for publication, you may indicate that here to bypass the “Comments to the Author” section, enter your conflict of interest statement in the “Confidential to Editor” section, and submit your "Accept" recommendation.

Reviewer #2: (No Response)

2. Is the manuscript technically sound, and do the data support the conclusions?

Reviewer #2: Yes

3. Has the statistical analysis been performed appropriately and rigorously? 

Reviewer #2: Yes

4. Have the authors made all data underlying the findings in their manuscript fully available?

Reviewer #2: Yes

5. Is the manuscript presented in an intelligible fashion and written in standard English?

Reviewer #2: Yes

6. Review Comments to the Author

Reviewer #2: General comments

I find that the manuscript has been improved and that my previous comments have been considered. Some more edits are needed, in my opinion. Please check thoroughly once more the grammar and that the descriptions of conformation (e.g. nose profile) are correct.

The breed/type names are a bit confusing for me, it is good that the authors explain at first mentioning that one breed/type is known under several names (e.g. Djallonké and West African Dwarf sheep), but it would be easier to follow the text throughout the manuscript if you would consistently stick to one of the names thereafter (IF they are truly referring to the same sheep type, otherwise please explain).

Specific comments

Line 28: do you mean among (or between) phytogeographical zones?

Line 29 + 31: I assume that with the precision of measurements in cm that was possible, it would be enough to give the mean with one decimal here, and the SE (is it SE – please clarify) with two decimals. (Also on line 200-201).

Line 55: seems instead of seemed?

Line 59: change ‘would have’ to something else like ‘is likely to have’

Line 60: could ‘assimilated to’ be removed here?

Line 62: change ‘with black forequarters’ to ‘, black forequarters’.

Line 63: rewrite e.g. as ‘Two sub-types of ……. differentiated by size have been distinguished …’

Line 65: Regroup sounds a bit odd here to me, do you mean include or something else (e.g. comprise)?

Line 70: Some of the pictures give the impression that the noses tend to be convex rather than concave? Please check this and make sure to change throughout the manuscript (including tables and figure texts) IF you would find out that you wrote this wrong.

Line 101: Consider changing ‘into’ to ‘in’.

Line 125: Rewrite ‘Thus, a total of 1240 ewes that were at least two years old and multiparous (at least two lambings)....’. I think farrowing is more commonly used for pigs (?).

Table 3: perhaps references would be a better word than authors on the third column head?

Line 216-217: Wouldn’t lamb be better to use than kid here (e.g. single-born lamb)

Line 229: Clarify – significant for what? For example, significantly contributing to discrimination between groups (?).

Table 7: It would be easier to read if you use the same number of decimals (e.g. 2) for all variables in the table, and adjust it so that the dots are beneath each other.

Table 8: Please explain more clearly what is above vs below the diagonal in the table.

Line 335: what is the color pie-brown, is that not also a bicolor/piebald?

Line 347 (and 335): are the Oudah and Bali-bali the same or different breeds/types?

Line 355-356: One decimal would be enough here I think, also for the means in Line 371-372.

Line 421: is assimilated the correct word here?

7. PLOS authors have the option to publish the peer review history of their article (what does this mean?). If published, this will include your full peer review and any attached files.

Reviewer #2: No

---

## [Author Response · Author response to Decision Letter 1]

9 Sep 2021

Additional Editor Comments:

PLOS_BeninSheep2

The manuscript is greatly improved. There are however several issues remaining.

1. We asked for the results to be put into a broader focus. The background you provided is all about sheep in Benin, but this can be scaled back and a broader geographic view taken. This makes the study more general, to students of sheep on the African continent and elsewhere. What is known about radiation or adaptive evolution of sheep breeds in other countries or in other parts of Africa?

Response: The results have been put into a broader focus as follows:

 Lines 379-382:” This study highlights a highly diverse sheep population in Benin, as in other African countries (e.g., Burkina Faso, Ivory Coast, Togo, and Nigeria), within which the distribution of individuals is affected by natural and also anthropogenic factors. Thus, the sheep subgroups observed in the different phytogeographic zones of Benin also exist in other African countries in similar or different environments [16,31,36,44].”

 Lines 413-417: “This calls for the development of new management strategies for sheep farming in Benin as well as in other African countries aiming to improve farm profitability by improving animal performance while preserving the diversity within the local sheep populations. In this way, sheep farming would overcome current and future challenges in production systems in Africa, including climate change and market demand.”

2. Improve the grammar and wording. Get an outsider to review the paper for you.

Examples of sentences that need improving 

Abstract: "Good knowledge…"

Line 50 "In West Africa, sheep populations are raised under harsh and diverse ecological conditions, which may have led to the evolution of…"

Line 65 "Djallonké sheep are assimilated to small-sized"?? better verb?

Line 79 drop "consistent"

Line 85 "To date, neither of these two characterization tools have not been covered in depth for of the Beninese sheep populations."????

There are several examples where extra words are used, that can be dropped. E.g. 510 "fully acknowledge"

Response: All the manuscript has been edited to ensure language and grammar accuracy be Editage (https://app.editage.com). The Certificate has been provided in the cover letter submitted to the editor. Some sentences have been improved as follows:

e.g. Line 15: “Knowledge of both the genetic diversity and…”

Lines 45-46: “In West Africa, sheep populations are raised under harsh and diverse ecological conditions, which may have led to the evolution of diversified adaptive traits for their survival [2,8].”

3. Indicate the origin of the map, software, database etc.

Response: The origin of the map given on the Figure 1 has been corrected (Source: Topographic Map IGN, 1992. Field Work, 2018-2020. WHANNOU Habib R. V. adapted from Adomou (2005)) and the software used to draw the map has been added in the tittle of the figure as follows: “Map of the vegetation zones and phytogeographic zones of Benin showing the 32 communes sampled to assess the morphological variability within the indigenous sheep population of Benin. The map was made using QGIS 3.8 [24].”

4. Figure legends should be extended. They should better describe the content of the figure.

Response: The legends have better described in the manuscript. E.g.: “Fig 1. Map of the vegetation zones and phytogeographic zones of Benin showing the 32 communes sampled to assess the morphological variability within the indigenous sheep population of Benin. The map was made using QGIS 3.8 [24].”

5. Tone downs statements of adaptive value of traits throughout manuscript.

a. For instance, in discussion (line 441) "These specific traits may allow them to withstand heat stress and to adapt"

Response: The statements of adaptive value traits have been tone down through the manuscript. E.g. Lines 399-402:” These specific traits allow them to reflect solar radiation better, and thus, are less prone to heat stress [45]. In addition, their long legs predispose them to travel long distances when searching for pastures. Moreover, their large height allows them to feed easily in tree and shrubs savannah pastures, which are predominant in these regions [22,23].”

#Journal Requirements:

Response: The reference list has been reviewed and some changes have been done. Citation errors have been corrected in the lines 61-64: “small narrow-erected ears, and a hairy short coat [14,17]. In contrast to ewes, rams are horned and have a heavy white or pied mane black forequarters and white hindquarters. Two sub-breeds of Djallonké sheep have been identified based on size [18,19]: the larger breed is found in the Sudanian zone, and the smaller breed in the Guinean zone further south [9,18].” 

A new reference has been cited in the tittle of Figure 1 as follows: “The map was made using QGIS 3.8 [24].”

#Specific comments

Line 28: do you mean among (or between) phytogeographical zones?

Response: This word has been corrected in the text in Line 26-27: “Univariate analyses indicated that all quantitative linear body measurements varied significantly (P<0.05) across the phytogeographic zones”

Line 29 + 31: I assume that with the precision of measurements in cm that was possible, it would be enough to give the mean with one decimal here, and the SE (is it SE - please clarify) with two decimals. (Also on line 200-201).

Response: For consistence, the mean values have been given with one decimal and the SE with two decimals throughout the manuscript.

Line 55: seems instead of seemed?

Response: This word has been corrected in the text in Line 54: “, also named West African Dwarf sheep, seems to originate....”

Line 59: change 'would have' to something else like 'is likely to have'

Response: This word group has been changed in the sentence in Line 59: “However, Djallonké sheep may have undergone …”

Line 60: could 'assimilated to' be removed here?

Response: This word group has been removed in Line 60: “Generally, Djallonké/WAD sheep are small-sized animals”

Line 62: change 'with black forequarters' to ', black forequarters'.

Response: This word group has been changed in Lines 61-62 as follows: “rams are horned and have a heavy white or pied mane black forequarters and white hindquarters. …”

Line 63: rewrite e.g. as 'Two sub-types of ……. differentiated by size have been distinguished …'

Response: This sentence has been changed in Lines 62-63: “Two sub-breeds of Djallonké sheep have been identified based on size [18,19] ….”

Line 65: Regroup sounds a bit odd here to me, do you mean include or something else (e.g. comprise)?

Response: This verb has been changed in Line 65: “Sahelian sheep include all long-legged sheep breeds ….”

Line 70: Some of the pictures give the impression that the noses tend to be convex rather than concave? Please check this and make sure to change throughout the manuscript (including tables and figure texts) IF you would find out that you wrote this wrong.

Response: Yes, it was a mistake. This word has been corrected throughout the manuscript.

Line 101: Consider changing 'into' to 'in'.

Response: This word has been corrected in the line 108:” This study was conducted in the 10 phytogeographic zones ….”

Line 125: Rewrite 'Thus, a total of 1240 ewes that were at least two years old and multiparous (at least two lambings) ....'. I think farrowing is more commonly used for pigs (?).

Response: This sentence has been rewritten and the words “farrowing” and “kids” has been corrected in the lines 127-128 as follows: “Thus, a total of 1240 ewes that were at least two years old and multiparous (at least two lambings) were randomly selected, described and phenotypically characterized.”

Table 3: perhaps references would be a better word than authors on the third column head?

Response: The word “Authors” has been corrected in the table 3.

Line 216-217: Wouldn't lamb be better to use than kid here (e.g. single-born lamb) 

Response: This word has been corrected in the lines 213-215: “Irrespective of the zone, single-born lambs were the most common. The highest percentages of twin-born lambs were recorded in the Oueme Valley, Pobe, and Zou zones, whereas the highest proportions of triplets and quadruplets were recorded in the Pobe zone. The percentage of multiple births appeared to ….”

Line 229: Clarify - significant for what? For example, significantly contributing to discrimination between groups (?).

Response: This has been clarified in lines 225-226 as follows:” … (i.e., 25 quantitative linear body traits and 14 morphological indices) included in the analysis significantly contribute to discrimination between the phytogeographic zones (P<0.0001).”

Table 7: It would be easier to read if you use the same number of decimals (e.g. 2) for all variables in the table, and adjust it so that the dots are beneath each other.

Response: This has been corrected in the table and the values has been adjusted to have dots beneath each other.

Table 8: Please explain more clearly what is above vs below the diagonal in the table.

Response: The values above vs below the diagonal in the table 8 has been explained below the table as follows: “Values above and/or below the diagonal represent the percentage of individuals from other phytogeographic zones present in the zone considered by the diagonal value.”

Line 335: what is the color pie-brown, is that not also a bicolor/piebald?

Response: Yes, it is also bicolor coat. 

Line 347 (and 335): are the Oudah and Bali-bali the same or different breeds/types?

 Response: No, Oudah and Bali-bali are Sahelian sheep but they are two different breeds.

Line 355-356: One decimal would be enough here I think, also for the means in Line 371-372.

Response: For consistence with the others part of the manuscript, the mean values have been given with one decimal and the SE with two decimals.

Line 421: is assimilated the correct word here?

Response: This verb has been changed in line 421: “Four sheep sub-populations were identified. Animals in the phytogeographic zones of Southern Benin could be identified as short-legged...”

---

## [Editor Report · Decision Letter 2]

1 Oct 2021

PONE-D-21-06521R2Morphological variability within the indigenous sheep population of BeninPLOS ONE

Dear Dr. DOSSA,

Thank you for submitting your manuscript to PLOS ONE. After careful consideration, we feel that it has merit but does not fully meet PLOS ONE’s publication criteria as it currently stands. Therefore, we invite you to submit a revised version of the manuscript that addresses the points raised during the review process.

The manuscript is greatly improved. There are two minor issues, that should only take a day to fix.

In the previous edit we requested you … [ down statements of adaptive value of traits throughout manuscript.]. this was not adhered to.

Line 452

“These specific traits allow them to reflect solar radiation better, and thus, are less prone to heat stress [45]. In addition, their long legs predispose them to travel long distances when searching for pastures. Moreover, their large height allows them to feed easily in tree and shrubs savannah pastures, which are predominant in these regions [22,23]”

Please rewrite this, remove the assertive tone and make this more nuanced. Please flank with caveats like “Several hypothesis about the adaptive value of these traits have been put forth. …. But confirmation of these hypotheses requires further study and remains inconclusive”

Line 484.

Is there a citation for this conclusion? “However, it appears that the indigenous sheep population of Benin has been subjected to very little selective breeding.” It is not a natural conclusion from your data. Suggest you remove this if no citation is available.

We look forward to receiving your revised manuscript.

Kind regards,

Arnar Palsson, Ph.D.

Academic Editor

PLOS ONE

Journal Requirements:

Additional Editor Comments (if provided):

The manuscript is greatly improved. There are two minor issues, that should only take a day to fix.

In the previous edit we requested you … [ down statements of adaptive value of traits throughout manuscript.]. this was not adhered to.

Line 452

“These specific traits allow them to reflect solar radiation better, and thus, are less prone to heat stress [45]. In addition, their long legs predispose them to travel long distances when searching for pastures. Moreover, their large height allows them to feed easily in tree and shrubs savannah pastures, which are predominant in these regions [22,23]”

Please rewrite this, remove the assertive tone and make this more nuanced. Please flank with caveats like “Several hypothesis about the adaptive value of these traits have been put forth. …. But confirmation of these hypotheses requires further study and remains inconclusive”

Line 484.

Is there a citation for this conclusion? “However, it appears that the indigenous sheep population of Benin has been subjected to very little selective breeding.” It is not a natural conclusion from your data. Suggest you remove this if no citation is available.
---

## [Author Response · Author response to Decision Letter 2]

5 Oct 2021

Additional Editor Comments (if provided):

Comment 1:

The manuscript is greatly improved. 

There are two minor issues, that should only take a day to fix.

In the previous edit we requested you … [ down statements of adaptive value of traits throughout manuscript.]. this was not adhered to.

Line 452 "These specific traits allow them to reflect solar radiation better, and thus, are less prone to heat stress [45]. In addition, their long legs predispose them to travel long distances when searching for pastures. Moreover, their large height allows them to feed easily in tree and shrubs savannah pastures, which are predominant in these regions [22,23]".

Please rewrite this, remove the assertive tone and make this more nuanced. Please flank with caveats like "Several hypothesis about the adaptive value of these traits have been put forth. …. But confirmation of these hypotheses requires further study and remains inconclusive".

Response: We sincerely thank the editor for his positive feedback on our revised submission. 

The statements of adaptive value traits have revised as suggested throughout the manuscript as follows:

• Lines 358-360: “This result is likely to confirm the effect of environmental factors on the morphology of sheep [40-43] and transhumance and management practices.”

• Lines 382-385: “The most important natural factors at the origin of the recorded sheep diversity across the 10 investigated phytogeographic zones might be climate-related factors (temperature, humidity, and/or vegetation cover), which affect the availability of feed resources and induce natural selection pressures.”

• Lines 399-404: “Several hypotheses about the adaptive value of these traits have been put forth. For instance, Gaughan et al. (2018) [45] argue that these specific traits might allow them to reflect solar radiation better, and thus, to be less prone to heat stress. In addition, according to these authors their long legs might predispose them to travel long distances when searching for pastures. Moreover, their large height might allow them to feed easily in tree and shrubs savannah pastures, which are predominant in these regions [22,23]. But confirmation of these hypotheses requires further study and remains inconclusive.”

Comment 2: 

Line 484. Is there a citation for this conclusion? "However, it appears that the indigenous sheep population of Benin has been subjected to very little selective breeding." It is not a natural conclusion from your data.

Suggest you remove this if no citation is available.

Response: This sentence has been removed 

#Journal Requirements:

Response: The reference list has been reviewed and some changes made from line 463 to line 591 to respond with Compliance with PLOS ONE’s style. 

Two new references [46,47] have been cited on line 415 and added to the list

---

## [Editor Report · Decision Letter 3]

6 Oct 2021

Morphological variability within the indigenous sheep population of Benin

PONE-D-21-06521R3

Dear Dr. DOSSA,

We’re pleased to inform you that your manuscript has been judged scientifically suitable for publication and will be formally accepted for publication once it meets all outstanding technical requirements.

Kind regards,

Arnar Palsson, Ph.D.

Academic Editor

PLOS ONE
---

## [Editor Report · Acceptance letter]

11 Oct 2021

PONE-D-21-06521R3 

Morphological variability within the indigenous sheep population of Benin 

Dear Dr. Dossa:

I'm pleased to inform you that your manuscript has been deemed suitable for publication in PLOS ONE. Congratulations! Your manuscript is now with our production department. 

Kind regards, 

on behalf of

Dr. Arnar Palsson 

Academic Editor

PLOS ONE